# Assessing Large Language Models (LLMs) as Foundational Recommenders: A Multi-Domain, Multi-Dataset Benchmark

## Abstract

Recent advances in large language models (LLMs) have raised the question of whether these models can serve as foundational recommenders across diverse domains. To systematically investigate this potential, we introduce RECBENCH-MD, a comprehensive benchmark for assessing the recommendation capabilities of LLMs from a multi-domain and multi-dataset perspective. Our benchmark encompasses 15 datasets spanning 10 domains, including e-commerce, entertainment, and social media, and evaluates 21 state-of-the-art LLMs under zero-resource, fine-tuning, and transfer-learning scenarios. Through extensive experiments, we reveal that (i) in-domain fine-tuning consistently delivers the strongest performance, (ii) cross-dataset transfer provides practical benefits in emerging recommendation contexts, and (iii) multi-domain training enhances the adaptability of LLM-based recommenders. These findings highlight both the opportunities and limitations of positioning LLMs as foundational recommenders. To support future research, we will publicly release all code and data[1].

## 1 Introduction

The rapid emergence of large language models (LLMs) has revolutionized various fields, ranging from natural language processing (NLP) (Touvron et al., 2023a; Reid et al., 2024) to computer vision and multimodal understanding (Kirillov et al., 2023; Li et al., 2024). Recently, their application in recommender systems has attracted considerable interest, as these models promise a unified framework capable of modeling user–item interactions through natural language (Wu et al., 2024a; Zhao et al., 2024; Bao et al., 2023b). Unlike traditional foundation models trained with domain-specific objectives, LLMs provide general-purpose reasoning and transferability, raising the possibility that they could serve as **foundational recommenders**, which should demonstrate broad zero-resource capabilities[2], enabling inference on and generalize to unseen datasets and even novel domains.

Despite this potential, our community still lacks systematic evidence on whether LLMs can indeed play this role. Most LLM-based recommendation studies focus on narrow settings, often restricted to a single dataset, domain, or recommendation paradigm, yielding partial and sometimes inconsistent conclusions. Consequently, there remains significant uncertainty regarding the generalizability of LLMs as cross-domain recommenders, requiring a systematic benchmark that spans multiple datasets, domains, training strategies, and recommendation approaches.

To fill this gap, we introduce RECBENCH-MD, the first multi-domain, multi-dataset benchmark explicitly designed to assess LLMs as foundational recommenders. Our taxonomy covers all key evaluation regimes (Figure 1), including settings from single-domain single-dataset evaluation (`Setting C`) to multi-domain multi-dataset training and transfer (`Setting H`). This unified framework allows us to systematically examine in-domain fine-tuning, cross-dataset transfer, and multi-domain training, capturing the full landscape of LLM recommendation scenarios.

---

[1] https://anonymous.4open.science/r/RecBench-MD

[2] In this paper, *zero-resource* refers to fine-tuning on some datasets and testing on unseen ones (cross-dataset), while *zero-shot* refers to testing without any fine-tuning.

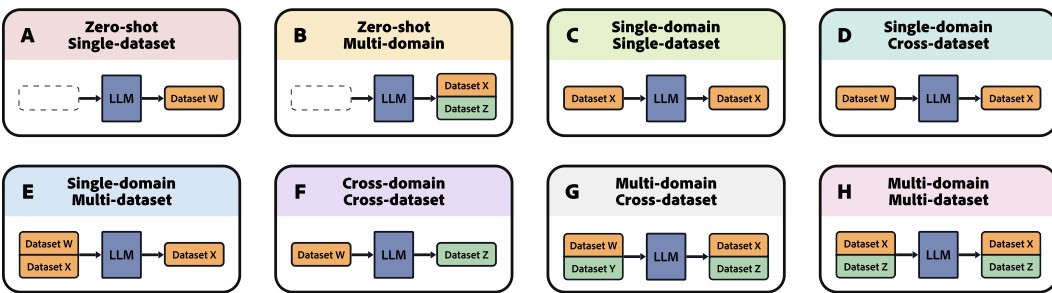

Figure 1: Illustrations of diverse recommendation settings, with colors denoting different domains.

Our benchmark is unprecedented in scope, evaluating 21 state-of-the-art LLMs across 15 datasets spanning 10 domains, including e-commerce, entertainment, and social media. We adopt both prompt-based ranking and embedding-based matching tasks, thus covering the principal paradigms in recommendation research. Beyond providing open-source code and datasets, we also ensure the benchmark's reproducibility and extensibility, enabling future researchers to conduct large-scale evaluations with minimal configuration effort.

Our benchmarking results reveals several key insights. **First,** larger models tend to benefit more from joint training on multiple datasets or domains, exhibiting stronger cross-domain generalization. **Second,** the degree of transferability across domains varies considerably, with a strong dependence on the characteristics of the source dataset. **Third,** cross-dataset transfer can serve as an effective model warm-up strategy in novel recommendation contexts, though it is challenging to exceed the performance upper bound established by fine-tuning on single or multiple datasets within the target domain. **Fourth,** by comparing with non–natural-language-based cross-domain recommendation baseline, i.e., RecBase Zhou et al. (2025), we observe that although LLMs suffer from low efficiency in tuning and inference, they achieve superior recommendation performance due to their rich prior knowledge.

## 2 RELATED WORK

**Existing Benchmarks.** Several benchmarks have been proposed to evaluate the recommendation capabilities for foundation models, including LLMRec (Liu et al., 2023), PromptRec (Wu et al., 2024b), and others (Zhang et al., 2021; Jiang et al., 2024; Liu et al., 2024a). However, as illustrated in Table 1, these benchmarks **i)** provide only a limited evaluation of recommendation settings, often focusing on a single approach. In addition, **ii)** the number of foundation models and datasets evaluated remains relatively small, resulting in an incomplete and fragmented performance landscape in this domain.

**Multi-domain Recommendation.** Traditional multi-domain recommendation methods predominantly rely on item-based or user-based knowledge transfer, using common items or shared user interactions to mitigate data sparsity and domain discrepancies (Guo et al., 2021; Chen et al., 2022; 2019). However, such approaches require explicit entity-level overlap between domains–a condition rarely met in real-world scenarios (Zhu et al., 2021; Zang et al., 2022). In contrast, text-based knowledge transfer leverages rich entity-side information, such as item descriptions and user profiles, used in diverse features (Chen et al., 2013; Gao et al., 2013). Borrowing the semantic comprehension and generation capabilities of foundation models, text-based methods boost cross-domain learning without the need for explicit entity alignment, i.e., non-overlap for users and items, thereby offering a more flexible and robust framework for transferring knowledge across heterogeneous domains.

**Foundation Models for Recommendation.** In recent years, integrating large language models (LLMs) into recommender systems has attracted significant academic and industrial interest. These integrations can be broadly classified into two paradigms (Wu et al., 2024a; Zhao et al., 2024; Bao et al., 2023b; Chen et al., 2024): LLM-FOR-RS and LLM-AS-RS. The LLM-FOR-RS paradigm enhances traditional recommenders via feature engineering or encoding techniques using LLMs (Wei et al., 2024; Liu et al., 2024b;c; 2025a; Wu et al., 2023; Zhou et al., 2025; Hu et al., 2024). In contrast, the LLM-AS-RS paradigm employs LLMs directly as recommenders (Ngo & Nguyen, 2024; Li

Table 1: Comparison of our RECBENCH-MD with existing benchmarks. "–" indicates that RecBole-CDR theoretically supports the corresponding feature, although no experimental results are provided.

| Benchmark
Year | Zhang et al.
2021 | OpenP5
2024 | LLMRec
2023 | PromptRec
2024b | Jiang et al.
2024 | RSBench
2024a | RecBole-CDR
2022 | RECBENCH
2025b | RECBENCH-MD
(ours) |
|---|---|---|---|---|---|---|---|---|---|
| **Scale** #Foundation Models | 4 | 2 | 7 | 4 | 7 | 1 | 0 | 17 | 21 |
| #Dataset | 1 | 10 | 1 | 3 | 4 | 3 | 3 | 5 | 15 |
| **Setting** Zero-shot | ✓ | × | ✓ | ✓ | × | × | × | ✓ | ✓ |
| Single-Dataset | ✓ | ✓ | ✓ | × | ✓ | ✓ | – | ✓ | ✓ |
| In-domain Cross-dataset | × | × | × | × | × | × | – | × | ✓ |
| In-domain Multi-dataset | × | × | × | × | × | × | – | × | ✓ |
| Cross-domain | × | × | × | ✓ | × | × | – | × | ✓ |
| Multi-domain | × | ✓ | × | × | × | × | – | × | ✓ |
| **Approach** Prompt-based | ✓ | ✓ | ✓ | ✓ | ✓ | ✓ | × | ✓ | ✓ |
| Embedding-based | × | × | × | × | × | × | – | × | ✓ |
| **Metric** Quality | ✓ | ✓ | ✓ | ✓ | ✓ | ✓ | – | ✓ | ✓ |
| Efficiency | × | × | × | × | × | × | × | ✓ | ✓ |

et al., 2023; Geng et al., 2022; Liu et al., 2024d). Studies have demonstrated its superior accuracy in contexts such as cold-start scenarios (Bao et al., 2023a) and in tasks requiring natural language understanding and generation (Luo et al., 2023; Wang et al., 2023; He et al., 2023).

# 3 PROPOSED BENCHMARK: RECBENCH-MD

## 3.1 RECOMMENDATION SETTINGS

In bottom-level text-based knowledge transfer, we can freely collect training data as long as: (i) each item is described by textual content, and (ii) each user is represented by their item consumption sequence. To systematically explore how cross-domain data influences target-domain recommendation performance, we propose a novel taxonomy comprising eight fine-tuning settings, as illustrated in Figure 1:

`Setting A` **(Zero-resource) Zero-shot Single-dataset.** The model is directly evaluated on a single dataset without any fine-tuning. This setting measures the model's intrinsic recommendation capability.

`Setting B` **(Zero-resource) Zero-shot Multi-domain.** A more comprehensive zero-shot evaluation: the model is tested on multiple datasets from different domains, and performance is averaged to assess generalization.

`Setting C` **Single-domain Single-dataset.** Fine-tuning and evaluation are performed on the same dataset. This setting reflects standard in-domain supervised learning.

`Setting D` **(Zero-resource) Single-domain Cross-dataset.** The model is fine-tuned on one or more datasets within a domain and evaluated on a different dataset from the same domain. It assesses domain-level generalization across datasets.

`Setting E` **Single-domain Multi-dataset.** Training and testing data are drawn from multiple datasets within the same domain, with potential overlap. This setting measures the benefit of aggregating in-domain data.

`Setting F` **(Zero-resource) Cross-domain Cross-dataset.** The model is fine-tuned on one domain and tested on a completely different one. This setting probes cross-domain transferability of recommendation knowledge.

`Setting G` **(Zero-resource) Multi-domain Cross-dataset.** Training and testing datasets come from overlapping but non-identical domains. This setting evaluates how auxiliary domain knowledge contributes to target performance when datasets do not overlap.

`Setting H` **Multi-domain Multi-dataset.** Both domains and datasets overlap between training and testing. This setting examines the upper-bound performance achievable via comprehensive domain and dataset fusion.

Table 2: Datasets evaluated or finetuned in our benchmark.

| Dataset | Domain | Symbol | Test set | | | Finetune set | | | Used Attributes |
|---------|--------|--------|----------|--|--|--------------|--|--|-----------------|
| | | | #Sample | #Item | #User | #Sample | #Item | #User | |
| H&M | Fashion | ⑪ H&M | 20,000 | 15,305 | 5,000 | 100,000 | 50,319 | 25,000 | detail_desc |
| MIND | News | 📖 MIND | 20,006 | 3,088 | 1,514 | 100,000 | 5,481 | 7,606 | title |
| MicroLens | Video | 📱 Micro. | 20,000 | 11,073 | 5,000 | 100,000 | 18,658 | 25,000 | title |
| Goodreads | Book | 📘 Good. | 20,009 | 12,984 | 1,736 | 100,005 | 40,322 | 8,604 | original_title |
| Amazon CDs | Music | 🎵 CDs | 20,003 | 15,568 | 4,930 | 100,003 | 55,428 | 24,618 | title |
| MovieLens | Movie | 🎬 Movie. | 20,008 | 4,300 | 2,251 | - | - | - | title |
| Yelp | Restaurant | 🍴 Yelp | 20,003 | 15,239 | 4,013 | - | - | - | name |
| Steam | Game | 🎮 Steam | 20,000 | 2,216 | 5,000 | - | - | - | game_name |
| Amazon Electronics | E-commerce | 🖥 Elec. | 20,002 | 11,045 | 5,431 | - | - | - | title |
| HotelRec | Hotel | 🏠 Hotel. | 20,002 | 17,295 | 5,437 | - | - | - | name, location |
| POG | Fashion | ⑪ POG | - | - | - | 100,002 | 15,846 | 15,734 | title_en |
| PENS | News | 📖 PENS | - | - | - | 100,007 | 9,053 | 8,542 | title |
| Netflix | Video | 📱 Netflix | - | - | - | 100,010 | 3,645 | 13,424 | title |
| Amazon Books | Book | 📘 Books | - | - | - | 100,002 | 28,471 | 25,139 | title |
| LastFM | Music | 🎵 Last.fm | - | - | - | 100,100 | 94,319 | 910 | track, artist |

## 3.2 RECOMMENDATION APPROACHES

We evaluate recommendation capabilities of foundation models with the pair-wise user–item click prediction task. It involves the estimation of the probability $\hat{y}$ that a user will interact positively with a candidate item. Therefore, the models will be trained by the binary cross-entropy (BCE) loss, formulated as:

$$\mathcal{L} = -\sum \left[ y \log \hat{y} + (1-y) \log(1-\hat{y}) \right],\tag{1}$$

where $y \in \{0, 1\}$ denotes the ground-truth label. Borrowing the idea from conventional recommendation, including matching-based and ranking-based models, we devise two recommendation approaches to calculate the click probabilities.

**Prompt-based Recommendation.** We concatenate the user sequence with the candidate item, i.e., in a **single-stream architecture**, where each item in the sequence and the candidate item are represented by their textual feature. Then, the entire user-item sequence will be in conjunction with a task-specific instruction (e.g., *"Will the user be interested in this item? Answer (Yes or No):"*). Next, the model is guided to predict specific output tokens (i.e., "Yes" or "No"), and their corresponding logits, $l_{\text{yes}}$ and $l_{\text{no}}$. Finally, the click probability $\hat{y}$ can be denoted as:

$$\hat{y} = \frac{e^{l_{\text{yes}}}}{e^{l_{\text{yes}}} + e^{l_{\text{no}}}}.\tag{2}$$

**Embedding-based Recommendation.** Following matching-based **two-tower paradigm**, here the foundation models are employed as user and item encoders learn their dense representations (embeddings) within a shared latent space. Specifically, we use the last token output embedding for user/item representation when the input is the user sequence or the candidate item. The click probability can be subsequently measured by the cosine similarity:

$$\hat{y} = \frac{\mathbf{u} \cdot \mathbf{t}}{\|\mathbf{u}\| \|\mathbf{t}\|},\tag{3}$$

where $\cdot$ denotes the dot product operation, $\|\|$ represents the L2 norm, and $\mathbf{u}$ and $\mathbf{t}$ are user and item representations.

## 4 EXPERIMENTAL SETUP

**Datasets.** To meaningfully probe foundation model capabilities in recommendation beyond prevalent single-dataset evaluations, a deliberately heterogeneous suite of 15 public datasets across 10 domains was assembled. This collection's scale and diversity (Table 2) are necessary to stress-test the central

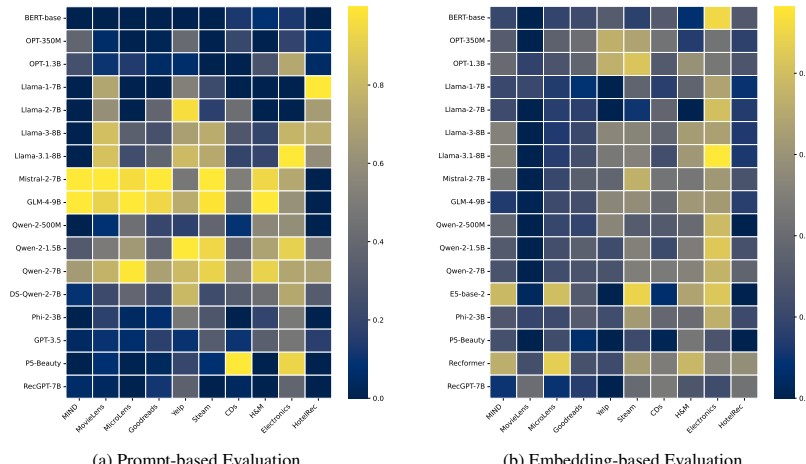

(a) Prompt-based Evaluation                    (b) Embedding-based Evaluation

Figure 2: Zero-shot evaluation. The light-colored (yellow) areas indicate better recommendation capabilities. Only AUC metric is reported due to the page limit.

premise of foundation model generalization across varied recommendation contexts, spanning high-volume consumer arenas (e.g., fashion, news) to specialized niches (e.g., games, hotels). The inherent heterogeneity manifesting in item taxonomies, user interaction dynamics, textual signal richness, and sparsity levels is instrumental, leveraged to transcend potentially idiosyncratic single-domain observations and evaluate genuine cross-domain. For each dataset, the fine-tuning set is randomly split into a training set and validation set in a 9:1 ratio.

**Foundation Models.** We collected 21 foundation models from different perspectives to evaluate their recommendation capabilities, including: $BERT_{base}$ (Kenton & Toutanova, 2019), $OPT_{350M}$ (Zhang et al., 2022), $OPT_{1B}$ (Zhang et al., 2022), $Llama-1_{7B}$ (Touvron et al., 2023a), $Llama-2_{7B}$ (Touvron et al., 2023b), $Llama-3_{8B}$ (Dubey et al., 2024), $Llama-3.1_{8B}$ (Meta AI, 2024), GPT-3.5 (OpenAI, 2023), $Qwen-2_{500M}$ (Yang et al., 2024), $Qwen-2_{1.5B}$ (Yang et al., 2024), $Qwen-2_{7B}$ (Yang et al., 2024), $GLM-4_{9B}$ (GLM et al., 2024), $Mistral-2_{7B}$ (Jiang et al., 2023), $DS-Qwen-2_{7B}$ (Bi et al., 2024), $E5_{base-v2}$ (Wang et al., 2022), $Phi-2_{3B}$ (Javaheripi et al., 2023), $RecGPT_{7B}$ (Ngo & Nguyen, 2024), $P5_{beauty}$ (Geng et al., 2022), and Recformer (Li et al., 2023). We present a comprehensive comparison across multiple dimensions: varying model sizes within the same organization (e.g., Qwen-2 series), different versions from the same organization (e.g., Llama series), models of similar size released in the same year by different organizations (e.g., THU's $GLM-4_{9B}$, Meta's $Llama-3_{8B}$, and Alibaba's $Qwen-2_{7B}$ in 2024), and models targeting different domains (e.g., the general foundation model Llama vs. the recommendation foundation model RecGPT). Specifically, the closed-source GPT-3.5 model from OpenAI supports only the prompt-based recommendation paradigm due to the unavailability of item and user embeddings. In contrast, models like Recformer and $E5_{base-v2}$, designed with a dual-tower architecture, can only be evaluated with the embedding-based paradigm.

**Evaluation Protocols.** Following common practice (Liu et al., 2025b), we evaluate recommendation performance using widely adopted metrics, including ranking metrics such as GAUC, NDCG, and MRR, as well as matching metrics like F1 and RECALL. However, **due to space limitations**, we present only the GAUC (shortly AUC) metric mostly. The full evaluation results will available on our webpage.

Additionally, we also design the Reciprocal Rank Average (RRA) metric to evaluate the contribution for each finetune set (used in Table 4). Specifically, we mark the top-K finetune set for each test set, and calculate the top-K RRA metric by:

$$\text{RRA@K} = \frac{1}{T} \sum_{i=1}^{T} \left( \mathbb{I}(r_i <= K) \cdot \frac{1}{r_i} \right), \tag{4}$$

where indicator function $\mathbb{I}(b) = 1$ if $b$ is true else 0, $\tag{5}$

Table 3: Performance comparison in single-domain fine-tuning scenario. We use cell background color to indicate different regimes, including `Setting B`, `Setting C`, and `Setting E`. Due to page limits, only AUC metric is reported here. More metrics can be found in Table 13, 14, and 15.

| Foundation Model | 👕 H&M | | 📰 MIND | | 📺 Micro. | | 📖 Good. | | 🎵 CDs | |
|---|---|---|---|---|---|---|---|---|---|---|
| | Prompt | Embedding | Prompt | Embedding | Prompt | Embedding | Prompt | Embedding | Prompt | Embedding |
| **Fine-tune set** | N/A | | | | | | | | | |
| BERT$_{base}$ | 0.5204 | 0.5167 | 0.4963 | 0.5263 | 0.4992 | 0.5305 | 0.4958 | 0.5160 | 0.5059 | 0.5139 |
| OPT$_{1B}$ | 0.5650 | 0.6370 | 0.5338 | 0.5510 | 0.5236 | 0.5447 | 0.5042 | 0.5258 | 0.4994 | 0.5137 |
| Llama-3$_{8B}$ | 0.5454 | 0.6487 | 0.4904 | 0.5666 | 0.5577 | 0.5218 | 0.5191 | 0.5150 | 0.5136 | 0.5162 |
| Mistral-2$_{7B}$ | 0.7166 | 0.6051 | 0.6300 | 0.5607 | 0.6579 | 0.5329 | 0.5718 | 0.5240 | 0.5230 | 0.5198 |
| Qwen-2$_{7B}$ | 0.7124 | 0.6201 | 0.5862 | 0.5347 | 0.6640 | 0.5391 | 0.5494 | 0.5190 | 0.5256 | 0.5212 |
| P5$_{beauty}$ | 0.7124 | 0.6201 | 0.4911 | 0.5948 | 0.5017 | 0.6423 | 0.5027 | 0.5186 | 0.5447 | 0.5218 |
| RecBase$_{base}$ | 0.5870 | - | 0.5508 | - | 0.5401 | - | 0.5029 | - | - | - |
| RecBase$_{large}$ | 0.6761 | - | 0.5442 | - | 0.5712 | - | 0.5329 | - | - | - |
| **Fine-tune set** | 👕 H&M | | 📰 MIND | | 📺 Micro. | | 📖 Good. | | 🎵 CDs | |
| BERT$_{base}$ | **0.8701** | 0.8359 | 0.7118 | 0.6837 | 0.8148 | **0.7671** | 0.5208 | 0.5738 | 0.6185 | **0.5909** |
| OPT$_{1B}$ | 0.8638 | 0.8163 | 0.7344 | 0.6788 | 0.8240 | 0.7545 | 0.5456 | 0.5754 | 0.6207 | 0.5713 |
| Llama-3$_{8B}$ | 0.8606 | 0.815 | 0.7120 | 0.6745 | 0.8295 | 0.7430 | **0.6799** | **0.5944** | 0.6267 | 0.5835 |
| RecBase$_{base}$ | 0.6261 | - | - | - | 0.5601 | - | - | - | - | - |
| **Fine-tune set** | 👕 H&M + 👕 POG | | 📰 MIND + 📰 PENS | | 📺 Micro. + 📺 Netflix | | 📖 Good. + 📖 Books | | 🎵 CDs + 🎵 Last.fm | |
| BERT$_{base}$ | 0.8009 | **0.8373** | 0.6834 | 0.6823 | 0.8112 | 0.7538 | 0.5169 | 0.5537 | 0.5790 | 0.5181 |
| OPT$_{1B}$ | 0.8132 | 0.7941 | **0.7350** | **0.6885** | 0.8237 | 0.7443 | 0.5966 | 0.5500 | 0.6151 | 0.5497 |
| Llama-3$_{8B}$ | 0.8216 | 0.8095 | **0.7350** | 0.6765 | **0.8368** | 0.7557 | 0.6325 | 0.5638 | **0.6353** | 0.5484 |

where $T$ is the number of the test datasets, $r_i$ is the rank of the model on the $i$-th dataset, $K$ is the rank threshold (e.g., $K = 5$).

**Implementation Details.** During data preprocessing, we standardized datasets of varying original sizes to comparable scales: the test set contains approximately 20,000 samples, while the fine-tuning set consists of around 100,000 samples. For each dataset, items were carefully curated to retain the most representative textual content features. User behavior sequences were truncated to a maximum length of 20; if a sequence exceeded this limit, only the most recent interactions were preserved.

We fine-tune models using LoRA (Hu et al., 2022) (Low-Rank Adaptation), a parameter-efficient strategy with rank 32 and alpha 128. The learning rate is set to 1e-4 across all experiments, using the Adam optimizer. An effective batch size of 32 is maintained via gradient accumulation, and early stopping is applied with a patience of 2. Models are built and evaluated using the Huggingface Transformers library (Wolf et al., 2019). For BERT$_{base}$, OPT$_{1B}$, and Llama-3$_{8B}$, the maximum sequence lengths are 512, 1024, and 1024, respectively, with precision set to float32 for BERT and bfloat16 for OPT$_{1B}$ and Llama-3$_{8B}$. To reduce fine-tuning overhead for embedding-based architectures, we freeze lower layers of OPT$_{1B}$ and Llama-3$_{8B}$, applying LoRA only to the top two layers. When fine-tuning on multiple datasets, early stopping is based on the average validation AUC across datasets. We will release the code, data, checkpoints, and documentation at our GitHub repository.

All the experiments are conducted on a single Nvidia A100 GPU device. Except for the zero-shot setting, all results are averaged over five runs, with statistically significant differences observed ($p < 0.05$).

## 5 FINDINGS AND DISCUSSIONS

In this section, we present a comprehensive analysis of experimental results evaluating the foundation model recommendation capabilities in diverse fine-tuning regimes and different evaluation tasks.[3]

---

[3]Due to space limits, more experimental results are provided in the appendix and supplementary material.

## 5.1 `Setting B` ZERO-SHOT MULTI-DOMAIN: PROMPT-BASED VS. EMBEDDING-BASED

Here, we investigate the zero-shot recommendation capabilities of various foundation models. For each dataset, we identify the maximum and minimum AUC values across all evaluated models in both paradigms (with the minimum constrained to 0.5) and normalize the results accordingly, as shown in Figure 2. Based on these findings, we make the following observations:

First, for almost all datasets, the prompt-based evaluation paradigm outperforms the embedding-based one, as it aligns more closely with the pre-training objectives of foundation models.

Second, under the prompt-based paradigm, three LLMs (Mistral-$2_{7B}$, GLM-$4_{9B}$, Qwen-$2_{7B}$) exhibit superior performance, possibly due to the inclusion of the collaborative signals during pre-training. In contrast, P5$_{beauty}$ performs well on two Amazon datasets (CDs and Electronics) but less favorably on others, likely because the used checkpoint was trained on the Amazon Beauty dataset, thereby modeling Amazon user interests.

Thirdly, in the embedding-based paradigm, performance differences among models are less pronounced. Notably, **smaller models** (such as BERT$_{base}$ and OPT$_{1B}$) **perform better under this setting than in the prompt-based paradigm**, whereas the embeddings of larger models appear less sensitive to similarity metrics, in line with the findings of (Freestone & Santu, 2024). Additionally, the matching-based language model E5$_{base-v2}$ and recommendation model Recformer also demonstrate strong performance, benefiting from the consistency between the evaluation and training paradigms.

## 5.2 SINGLE-DOMAIN FINE-TUNING: `Setting C` VS. `Setting E`

Here, we study the single-domain fine-tuning recommendation scenario. We mainly select three foundation models, i.e., BERT$_{base}$, OPT$_{1B}$, and Llama-$3_{8B}$, of three distinct model size for the evaluation. From results displayed in Table 3, we can make the following observations:

First, compared to zero-shot baselines (`Setting B`), **domain-specific fine-tuning strategies** (`Setting C` and `Setting E`) consistently **achieve superior performance** on both prompt-based and embedding-based paradigm. This is primarily because large models have acquired domain-specific collaborative knowledge through fine-tuning.

Second, **fine-tuning with a single-domain single-dataset** setting (`Setting C`) **yields more stable performance** than the cross-dataset variant (`Setting E`), even within the same domain. This is likely due to optimization conflicts between datasets, as observed on Goodreads and H&M, where `Setting E` underperforms compared to `Setting C`.

Third, **large-scale foundation models** (e.g., Llama-$3_{8B}$) **achieve the best performance** under the `Setting E`, as their pretraining enables a broad understanding of general textual knowledge across domains, allowing them to effectively extract and generalize useful patterns from auxiliary datasets to the target dataset. In contrast, smaller models such as BERT$_{base}$ are less suited for `Setting E`, as they struggle to abstract transferable patterns even from datasets within the same domain, leading to limited performance gains.

## 5.3 FOUNDATIONAL RECOMMENDERS: LLM V.S. RECBASE

RecBase Zhou et al. (2025) is a pretrain-from-scratch foundational recommender that leverages cross-domain user sequences and discrete semantic techniques (i.e., TIGER Rajput et al. (2023)) to generate semantic identifiers for items. This approach encodes cross-domain knowledge via global discretized identifiers rather than natural language, resulting in a compact and efficient model with zero-resource recommendation capability. As shown in Table 3 (`Setting B` and `Setting C` settings), RecBase demonstrates promising performance. Nevertheless, LLMs, with their rich priors, consistently outperform RecBase, albeit at the cost of much larger model sizes and lower efficiency in training and inference.

## 5.4 CROSS-DATASET FINE-TUNING: `Setting D` VS. `Setting F`

Here, we study effect of the cross-dataset fine-tuning, including single-domain and cross-domain scenario. The experiments are conducted across two foundation models: BERT$_{base}$ and Llama-$3_{8B}$.

Table 4: Performance comparison in cross-dataset fine-tuning scenario. We use cell background color to indicate different regimes, including `Setting B`, `Setting C`, `Setting D`, and `Setting F`. We mark the top-5 rank finetune set for each test set and bold the best. We use **red** color to indicate the result **inferior** to the zero-shot one or **small** than 0.5. Due to page limits, only AUC metric is reported here. More metrics can be found in Table 13, 14, and 15.

| | H&M | MIND | Micro. | Good. | CDs | Movie. | Yelp | Steam | Elec. | Hotel. | RRA@5 |
|---|---|---|---|---|---|---|---|---|---|---|---|
| **Foundation Model: BERT$_{base}$** | | | | | | | | | | | |
| N/A | 0.5204 | 0.4963 | 0.4992 | 0.4958 | 0.5059 | 0.4934 | 0.4914 | 0.5002 | 0.5037 | 0.4955 | - |
| H&M | **0.8701** (1) | 0.5496 (4) | 0.5692 (3) | **0.5282** (1) | 0.5103 (3) | 0.5127 (4) | 0.4961 | 0.7291 (3) | 0.5304 (5) | 0.4869 | **0.3833** (1) |
| MIND | 0.6750 (3) | **0.7118** (1) | 0.5877 (2) | 0.5255 (4) | 0.5128 (2) | 0.4932 | 0.5024 (5) | 0.7184 (4) | 0.5306 (2) | 0.4847 | 0.3533 (3) |
| Micro. | 0.6661 (4) | 0.5841 (3) | **0.8148** (1) | 0.5097 | 0.5093 (5) | 0.5150 (3) | 0.4864 | **0.7393** (1) | 0.5004 | 0.4807 | 0.3117 (4) |
| Good. | 0.6218 | 0.5081 | 0.5239 | 0.5208 (5) | 0.4992 | 0.4957 | 0.5105 (4) | 0.6220 | 0.5168 | 0.4952 (4) | 0.0700 (8) |
| CDs | 0.6464 (5) | 0.5053 | 0.5152 | 0.5139 | **0.6185** (1) | 0.5503 (2) | **0.5356** (1) | 0.4794 | 0.5076 | **0.5216** (1) | 0.3700 (2) |
| POG | 0.6222 | 0.5153 | 0.5470 | 0.5138 | 0.4989 | 0.4953 | 0.4913 | 0.6171 | 0.5291 (4) | 0.4914 (5) | 0.0450 (10) |
| PENS | 0.6872 (2) | 0.6203 (2) | 0.5554 (5) | 0.5165 | 0.5051 | 0.5069 | 0.4987 | 0.7311 (1) | 0.5218 | 0.4900 | 0.1700 (7) |
| Netflix | 0.6191 | 0.5396 (5) | 0.5328 | 0.5080 | 0.5097 (4) | **0.5656** (1) | 0.5117 (2) | 0.6954 (5) | 0.5255 (5) | 0.5077 (2) | 0.2683 (5) |
| Books | 0.6108 | 0.5108 | 0.5295 | 0.5264 (2) | 0.5089 | 0.5119 (5) | 0.5155 (2) | 0.6191 | **0.5313** (1) | 0.4957 (3) | 0.2533 (6) |
| Last.fm | 0.6279 | 0.5127 | 0.5645 (4) | 0.5263 (3) | 0.5023 | 0.4855 | 0.4773 | 0.6699 | 0.5231 | 0.4769 | 0.0583 (9) |
| **Foundation Model: Llama-3$_{8B}$** | | | | | | | | | | | |
| N/A | 0.5267 | 0.4904 | 0.6412 | 0.5577 | 0.5191 | 0.7690 | 0.5136 | 0.5454 | 0.5223 | 0.5342 | - |
| H&M | **0.8606** (1) | 0.5693 (5) | 0.6758 (2) | 0.6116 (2) | 0.5268 (5) | 0.6818 (4) | 0.4911 | 0.9193 (2) | 0.5469 (4) | 0.5116 | 0.3400 (2) |
| MIND | 0.6599 | **0.7120** (1) | 0.6104 (5) | 0.5279 | 0.5176 | 0.5235 | 0.4808 | 0.8033 | 0.5284 | 0.5008 | 0.1200 (9) |
| Micro. | 0.7504 (3) | 0.6331 (3) | **0.8295** (1) | 0.5829 | 0.5239 | 0.6703 | 0.5165 | 0.8953 (4) | 0.5250 | 0.4900 | 0.1917 (7) |
| Good. | 0.6592 | 0.5249 | 0.5731 | **0.6799** (1) | 0.5334 (4) | 0.6550 | 0.5301 (3) | 0.8795 (5) | 0.6051 (3) | 0.5320 (4) | 0.2367 (5) |
| CDs | 0.6262 | 0.5019 | 0.5385 | 0.5840 (5) | **0.6267** (1) | 0.7201 (2) | **0.5939** (1) | 0.6427 | **0.6410** (1) | 0.6057 (3) | **0.4033** (1) |
| POG | 0.5922 | 0.4912 | 0.5175 | 0.5068 | 0.5011 | 0.5354 | 0.5147 (4) | 0.5684 | 0.4896 | 0.5142 (5) | 0.0450 (10) |
| PENS | 0.7517 (2) | 0.6823 (2) | 0.6675 (3) | 0.5670 | 0.5191 | 0.6311 | 0.5134 (5) | 0.8979 (3) | 0.5353 | 0.4860 | 0.1867 (8) |
| Netflix | 0.6655 (5) | 0.4844 | 0.5634 | 0.5694 | 0.5355 (3) | **0.7422** (1) | 0.4958 | 0.8547 | 0.5947 (5) | 0.6104 (2) | 0.2367 (5) |
| Books | 0.5750 | 0.5040 | 0.5322 | 0.5876 (3) | 0.5600 (2) | 0.6935 (3) | 0.5727 (2) | 0.6373 | 0.6267 (2) | **0.6149** (1) | 0.2833 (3) |
| Last.fm | 0.7444 (4) | 0.5807 (4) | 0.6567 (4) | 0.5871 (4) | 0.5045 | 0.6736 (5) | 0.4818 | **0.9363** (1) | 0.5414 (5) | 0.5125 | 0.2400 (4) |

The foundation model will be firstly fine-tuned with one single dataset and then evaluated over 10 test datasets. We design a RRA metric (Equation 4) to evaluate the usefulness of each finetune set.

Based Table 4, we can make the following observations:

First, **cross-dataset fine-tuning** generally improves recommendation performance, but it **may also introduce negative effects** on the target dataset in some cases (as indicated by the red-highlighted results in the table). Notably, the ✗ Yelp and ⌂ Hotel. datasets exhibit a higher likelihood of such degradation, possibly due to domain gaps and mismatches between the test sets and finetune sets. Moreover, for the Llama-3$_{8B}$ model, ⌷ Micro. and ⊞ Movie. also demonstrate performance degradation under cross-dataset finetuning. Interestingly, these two datasets are where Llama-3$_{8B}$ achieves the highest zero-shot performance among the 10 test sets. This suggests that Llama-3$_{8B}$ likely encountered collaborative signals related to these domains during pretraining, allowing it to effectively capture user interests for video-based recommendations even without additional tuning.

Second, **single-domain cross-dataset finetuning is not always more effective than cross-domain finetuning**. While it intuitively makes sense that user interests are easier to model within the same domain–supported by results in news (▣ MIND–▣ PENS) and books (▤ Good.–▤ Books)–this trend does not hold for movies, music, and fashion: their `Setting D` results did not even rank in the top five. A possible reason is that ▣ MIND and ▣ PENS both originate from Microsoft, and Amazon is the source of ▤ Books as well as the parent company of Goodreads.com, suggesting these dataset pairs may share more similar distributions.

Third, **dataset quality varies, but its effectiveness also depends heavily on the capacity of the pretrained model**. For instance, finetuning on ▤ Good. with BERT$_{base}$ (`Setting C`) ranks only fifth, while using Llama-3$_{8B}$ lifts it to first. This may be because Goodreads relies on book titles as content features, which are poorly represented in smaller models' pretraining corpora. In contrast, Llama-

Table 5: Performance comparison in multi-domain fine-tuning scenario. We use cell background color to indicate different settings, including `Setting B`, `Setting G`, and `Setting H`. Due to page limits, only AUC metric is reported here. More metrics can be found in Table 13, 14, and 15.

| Model | 👕 H&M | 📖 MIND | 📺 Micro. | 📖 Good. | 🎵 CDs | 📽 Movie. | 🍴 Yelp | 🎮 Steam | 💻 Elec. | 🏠 Hotel. |
|---|---|---|---|---|---|---|---|---|---|---|
| BERT_base | 0.5204 | 0.4963 | 0.4992 | 0.4958 | 0.5059 | 0.4934 | 0.4914 | 0.5002 | 0.5037 | **0.4955** |
| | 0.6607 | 0.6004 | 0.5711 | 0.5213 | 0.5092 | **0.5675** | **0.5218** | 0.6189 | **0.5091** | 0.4795 |
| | **0.8569** | **0.7004** | **0.8019** | **0.5648** | **0.5653** | 0.5106 | 0.5093 | **0.6817** | 0.5039 | 0.4869 |
| OPT_1B | 0.5650 | 0.5338 | 0.5236 | 0.5042 | 0.4994 | 0.5174 | **0.5026** | 0.3825 | 0.5205 | 0.5026 |
| | 0.7002 | 0.5996 | 0.6165 | 0.5189 | 0.5181 | **0.6156** | 0.4853 | **0.7665** | **0.5446** | **0.5037** |
| | **0.8658** | **0.7259** | **0.8132** | **0.5374** | **0.6220** | 0.5813 | 0.5014 | 0.6959 | 0.5248 | 0.4951 |
| Llama-3_8B | 0.7690 | 0.4904 | 0.6412 | 0.5577 | 0.5191 | 0.5267 | 0.5136 | 0.5454 | 0.5223 | 0.5342 |
| | 0.7295 | 0.6732 | 0.6223 | 0.5864 | 0.5626 | **0.7203** | 0.5764 | **0.7828** | **0.6296** | **0.5806** |
| | **0.8524** | **0.7206** | **0.8235** | **0.6660** | **0.6281** | 0.6683 | **0.5846** | 0.7042 | 0.6139 | 0.5318 |

$3_{8B}$ better understands textual content, leading to more robust item representations and improved user modeling. According to the Top-5 RRA results, 🎵 CDs and 👕 H&M offer the strongest transferability, while 👕 POG performs the weakest. Additionally, 📖 Good. and 🎵 Last.fm show large performance gains when switching from BERT_base to Llama-$3_{8B}$, suggesting complex content features paired with highly transferable user interests. On the other hand, 📖 MIND and 📺 Micro. show ranking drops, indicating their simpler content may already be sufficiently modeled by smaller models, but their user behavior patterns are less suitable for cross-dataset transfer. Finally, although 📖 Books and 🎵 Last.fm do not have corresponding test sets, their finetuned models still rank in the top five under Llama-$3_{8B}$, suggesting strong generalization capability across domains.

## 5.5 MULTI-DOMAIN FINE-TUNING: `Setting G` VS. `Setting H`

We investigate the impact of multi-domain fine-tuning, focusing on two key settings: multi-domain cross-dataset ( `Setting G` ) and multi-domain multi-dataset ( `Setting H` ). In the `Setting G` setting, foundation models are fine-tuned using datasets from domains different from the test sets, specifically: 👕 POG, 📖 PENS, 📺 Netflix, 📖 Books, and 🎵 Last.fm. In contrast, the `Setting H` setting involves fine-tuning on datasets that share domains with the test sets but do not include overlapping data, namely: 👕 H&M, 📖 MIND, 📺 Micro., 📖 Good., and 🎵 CDs. From the results illustrated in Table 5, we can make the following observations:

First, `Setting H` achieves the best performance on 👕 H&M, 📖 MIND, 📺 Micro., 📖 Good., and 🎵 CDs across all three foundation models, as it is fine-tuned directly on these datasets and thus captures domain-specific knowledge effectively. Second, although `Setting G` uses different datasets for fine-tuning, it consistently outperforms the zero-shot setting ( `Setting B` ), highlighting **the generalization benefits of multi-domain training with diverse user behavior patterns**. Third, while `Setting H` serves as an upper bound, the performance gap between `Setting H` and `Setting G` narrows with larger models–for instance, the improvement on HM drops from 30.0% (BERT_base) to 16.8% (Llama-$3_{8B}$), suggesting that large models fine-tuned on cross-domain data can better handle zero-resource scenarios. Finally, due to its reliance on specific data distributions, `Setting H` underperforms `Setting G` on five other datasets (📽 Movie., 🍴 Yelp, 🎮 Steam, 💻 Elec., 🏠 Hotel.), underscoring the fairness and robustness of the multi-domain cross-dataset setting.

## 6 CONCLUSION

We have introduced RECBENCH-MD, a novel and comprehensive benchmark designed to evaluate the recommendation capabilities of foundation models across a wide range of datasets and domains. Our thorough analysis of 19 foundation models across 15 datasets and 10 domains provides crucial insights into their performance in recommendation tasks. The findings demonstrate the substantial advantages of cross-dataset transfer learning and multi-domain training in improving the adaptability of foundation models. We expect that these insights, along with the valuable resources provided, will drive future advancements in the development of recommendation foundation models, offering a strong foundation for continued research and innovation in this field.

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

## A    LIMITATIONS

In this study, we assess the recommendation capabilities of foundation models on two of the most prevalent tasks: prompt-based approaches (similar to CTR models) and embedding-based approaches (akin to matching models), from a multi-dataset, multi-domain perspective. Nonetheless, our current evaluation does not encompass sequential recommendation, which represents a crucial area for future development and enhancement.

## B    BROADER IMPACTS

Our benchmark offers a comprehensive and scalable framework for evaluating foundation models in zero-resource, multi-dataset, multi-domain recommendation scenarios, thereby promoting more systematic and reproducible research. It establishes a solid foundation for ongoing research and innovation in this field. Furthermore, the benchmark facilitates cross-domain fine-tuning, extending its benefits to other areas such as natural language processing.

## C  TECHNICAL APPENDICES

### C.1  EFFICIENCY ANALYSIS

Table 6: Efficiency analysis of various models on CPU and GPU on the prompt-based approach.

| Model | CPU (ms) | GPU (ms) |
|---|---|---|
| BERT-base | 53.26 | 11.08 |
| OPT-350M | 332.34 | 14.99 |
| OPT-1B | 1 140.00 | 16.38 |
| LLaMA-1-7B | 3 170.00 | 71.10 |
| LLaMA-2-7B | 6 200.00 | 71.90 |
| LLaMA-3-8B | 6 800.00 | 65.11 |
| LLaMA-3.1-8B | 6 580.00 | 66.39 |
| Mistral-7B | 7 680.00 | 76.14 |
| GLM-4-9B | 9 690.00 | 83.38 |
| Qwen-2-0.5B | 543.73 | 34.89 |
| Qwen-2-1.5B | 1 420.00 | 40.01 |
| Qwen-2-7B | 6 150.00 | 70.47 |
| DeepSeek-Qwen-2-7B | 7 520.00 | 61.60 |
| Phi-2-3B | 2 100.00 | 61.58 |
| RecGPT-7B | 7 160.00 | 54.34 |
| P5-Beauty | 74.11 | 12.30 |

Here, we report the time required for a single forward pass per model on the prompt-based approach.[4] As demonstrated in Table 6, larger models consistently incur higher latency on both CPU and GPU devices, raising concerns regarding the inference efficiency of LLMs. Techniques such as batch decoding—which accelerates inference by processing multiple items within a single input sequence and decoding them in parallel—represent a promising direction for improvement.

### C.2  IMPACT OF SAMPLE SIZE

Here, we study the impact of sample size. The experiments are conducted on `Setting C`, i.e., single-domain, single-dataset. As reported in Table 7, we can observe that, as the number of training samples increases, the performance of Llama-$3_{8B}$ improves. This is because each user is represented by a sequence of item interactions, and with more item occurrences (i.e., more training samples), the item embeddings are trained more thoroughly and become more robust.

Table 7: Ablation studies on sample size.

| Name | Train | Test | #Item | #User | Llama-$3_{8B}$ |
|---|---|---|---|---|---|
| CDs-small | 50,000 | 20,000 | 72,796 | 12,355 | 0.6071 |
| CDs | 100,000 | 20,000 | 113,671 | 24,602 | 0.6268 |
| CDs-large | 150,000 | 20,000 | 146,604 | 36,916 | 0.6300 |

### C.3  WHY RRA METRIC WORKS BETTER

To further clarify how RRA metric works better, here we append an additional "Avg" column to Table 4, which reports the average GAUC across the 10 test datasets for each finetune set. From

---

[4]Since embedding-based approach can be accelerated by similarity search techniques such as product quantization, evaluating the latency for a single user–item pair cannot reflect the real industrial scenario.

Table 8: Performance comparison in cross-dataset fine-tuning scenario. Experiments are conducted on the BERT$_{\text{base}}$ model. All the metrics are the same with Table 4.

| | H&M | MIND | Micro. | Good. | CDs | Movie. | Yelp | Steam | Elec. | Hotel. | RRA@5 | Avg |
|---|---|---|---|---|---|---|---|---|---|---|---|---|
| H&M | **0.8701** (1) | 0.5496 (4) | 0.5692 (3) | **0.5282** (1) | 0.5103 (3) | 0.5127 (4) | 0.4961 | 0.7291 (3) | 0.5304 (3) | 0.4869 | **0.3833** (1) | 0.5783 (2) |
| MIND | 0.6750 (3) | **0.7118** (1) | 0.5877 (2) | 0.5255 (4) | 0.5128 (2) | 0.4932 | 0.5024 (5) | 0.7184 (4) | 0.5306 (2) | 0.4847 | 0.3533 (3) | 0.5742 (3) |
| Micro. | 0.6661 (4) | 0.5841 (3) | **0.8148** (1) | 0.5097 | 0.5093 (5) | 0.5150 (3) | 0.4864 | **0.7393** (1) | 0.5004 | 0.4807 | 0.3117 (4) | **0.5806** (1) |
| Good. | 0.6218 | 0.5081 | 0.5239 | 0.5208 (5) | 0.4992 | 0.4957 | 0.5105 (4) | 0.6220 | 0.5168 | 0.4952 (4) | 0.0700 (8) | 0.5314 (10) |
| CDs | 0.6464 (5) | 0.5053 | 0.5152 | 0.5139 | **0.6185** (1) | 0.5503 (2) | **0.5356** (1) | 0.4794 | 0.5076 | **0.5216** (1) | 0.3700 (2) | 0.5394 (6) |
| POG | 0.6222 | 0.5153 | 0.5470 | 0.5138 | 0.4989 | 0.4953 | 0.4913 | 0.6171 | 0.5291 (4) | 0.4914 (5) | 0.0450 (10) | 0.5321 (9) |
| PENS | 0.6872 (2) | 0.6203 (2) | 0.5554 (5) | 0.5165 | 0.5051 | 0.5069 | 0.4987 | 0.7311 (2) | 0.5218 | 0.4900 | 0.1700 (7) | 0.5633 (4) |
| Netflix | 0.6191 | 0.5396 (5) | 0.5328 | 0.5080 | 0.5097 (4) | **0.5656** (1) | 0.5117 (3) | 0.6954 (5) | 0.5255 (5) | 0.5077 (2) | 0.2683 (5) | 0.5515 (5) |
| Books | 0.6108 | 0.5108 | 0.5295 | 0.5264 (2) | 0.5089 | 0.5119 (5) | 0.5155 (2) | 0.6191 | **0.5313** (1) | 0.4957 (3) | 0.2533 (6) | 0.5360 (8) |
| Last.fm | 0.6279 | 0.5127 | 0.5645 (4) | 0.5263 (3) | 0.5023 | 0.4855 | 0.4773 | 0.6699 | 0.5231 | 0.4769 | 0.0583 (9) | 0.5366 (7) |

Table 9: Impact of precision differences. Experiments are conducted on the H&M dataset.

| | Zero-shot ( Setting B ) | Fine-tune ( Setting C ) |
|---|---|---|
| BERT (float32) | 0.5204 | 0.8701 |
| BERT (bf16) | 0.5210 | 0.8688 |
| Llama3-8B (float32) | 0.5444 | 0.8598 |
| Llama3-8B (bf16) | 0.5454 | 0.8606 |

Table 8, we can clearly compare the difference between "RRA" and "Avg", that "Avg" is more likely to be biased. We highlight two illustrative cases:

1. MicroLens rises from Rank 4 (RRA) to Rank 1 (AVG): The main driver of this jump is its exceptional performance on the MicroLens test set itself, where it achieves 0.8148 GAUC – significantly outperforming the second-best (MIND with 0.5877) by nearly 25 points. Since MicroLens is one of the 10 test datasets, its high GAUC skews the average and inflates the ranking of the MicroLens finetune set.

2. CDs drops from Rank 2 (RRA) to Rank 6 (AVG): The CDs finetune set achieves the top GAUC on three different test datasets—more than any other finetune set. For example, on HotelRec, which is considered one of the most challenging test datasets, the CDs finetune set achieves 0.5216, making it one of only two finetune sets to surpass 0.5 GAUC. However, this strength on difficult datasets is not adequately reflected in the average metric, which treats all test datasets equally.

Therefore, we can summarize the key motivation for RRA: although we control for dataset size across all finetune and test sets, the intrinsic difficulty of each dataset still varies. Simple averaging tends to favor finetune sets that perform well on easier datasets, while RRA explicitly rewards robustness across a range of diverse and potentially hard distributions. Thus, RRA provides a fairer and more nuanced measure of generalization.

## C.4   IMPACT OF PRECISION DIFFERENCES FOR DIFFERENT MODELS

For efficiency reasons (for in total over 900 experiments), we used bfloat16 precision for all models larger than 7B, which is standard practice in large-scale training and inference. To assess its impact, here we study on the H&M dataset (Table 9) and found that the difference in predictive performance between float32 and bfloat16 was negligible in our recommendation tasks.

## C.5   PROMPT ANALYSIS

We have provided the used prompt in our code repository. To further address your concern regarding the prompt templates, we conducted experiments using the prompts depicted in Table 10.

Specifically, P1 is a concise prompt; P2 is more detailed; and P2 (1-shot) follows the in-context learning paradigm by including a demonstration example. The average GAUC scores across five

Table 10: List of prompts used in our benchmark.

| Version | Prompt |
|---|---|
| P1 (zero-shot) | You are a recommender. Please respond "YES" or "NO" to represent whether this user is interested in this item.
User history sequence: `[object Object]`.
Candidate item: `[object Object]`.
Answer (Yes/No): |
| P2 (zero-shot) | You are a recommender. I will provide user behavior sequence and a candidate item. Please respond "YES" or "NO" only. You are not allowed to give any explanation or note. Now, your role formally begins. Any other information should not disturb you.
User history sequence: `[object Object]`.
Candidate item: `[object Object]`.
Answer (Yes/No): |
| P2 (1-shot) | You are a recommender … (*example omitted*) …
User history sequence: `[object Object]`.
Candidate item: `[object Object]`.
Answer (Yes/No): |
| P2 (2-shot) | *Omitted due to space constraints.* |
| P2 (5-shot) | *Omitted due to space constraints.* |

Table 11: Impact of prompts. Experiments are conducted on the Qwen-3$_{8B}$ model.

| Setting | 📖 MIND | 📺 Micro. | 📘 Good. | 🎵 CDs | 🎁 H&M |
|---|---|---|---|---|---|
| P1 (zero-shot) | 0.5847 | 0.6543 | **0.5439** | 0.5180 | 0.6499 |
| P2 (zero-shot) | 0.6036 | **0.6618** | 0.5371 | 0.5175 | **0.6678** |
| P2 (1-shot) | 0.6097 (48) | 0.6254 (43) | 0.5356 (56) | 0.5056 (38) | 0.6178 (16) |
| P2 (2-shot) | **0.6354** (50) | 0.6536 (56) | 0.5360 (60) | 0.5150 (69) | 0.6105 (146) |
| P2 (5-shot) | 0.6208 (52) | 0.6565 (71) | 0.5429 (28) | **0.5188** (49) | 0.6176 (124) |

datasets (MIND, MicroLens, Goodreads, CDs, and H&M) using the Qwen-3$_{8B}$ model under these prompts are summarized in Table 11, from which we can observe that:

First, overall, for zero-shot performance, P2 offers only minor improvements over P1. Therefore, in our experiments, we used P1 for smaller models (size < 7B) with shorter input windows to conserve input tokens and allow for longer user sequences. For larger models with longer input windows, we used P2.

Second, the few-shot experiments were conducted three times, with each run using different examples randomly selected for in-context demonstrations. Compared to zero-shot performance, we observed that few-shot prompting does not always lead to improved results and can, in some cases, cause a significant drop in performance, as seen on the H&M dataset.

## C.6    IMPACT OF FINE-TUNING DATASET ORDER IN MULTI-DOMAIN RECOMMENDATION

Previously, Table 5 compares three evaluation strategies: `Setting B`, `Setting G`, and `Setting H`. Both `Setting G` and `Setting H` involve training on a mixture of all available fine-tune sets. In contrast, we analyze a sequential fine-tuning strategy here, focusing on how to select datasets and determine fine-tuning order. Given the combinatorial complexity of all five datasets ($A_5^5$), we restrict our analysis to pairs of fine-tune sets. Based on the results from Table 12, we can observe that:

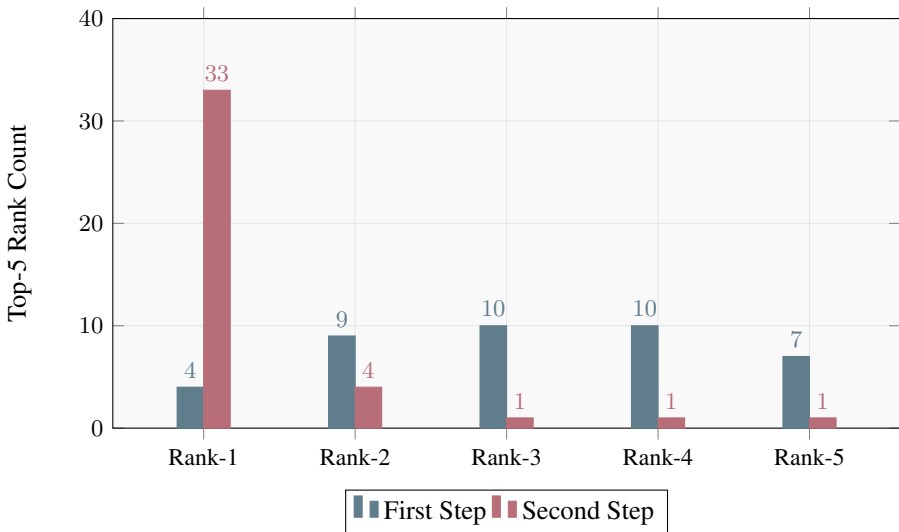

Figure 3: Impact of fine-tuning dataset order in multi-domain fine-tuning. The statistics are derived from Table 12. For each test set, we select the top-5 AUC scores from the 25 multi-domain combinations (excluding single-dataset fine-tuning results). For each top-5 result (e.g., 0.6687 on ▦ MIND), where fine-tuning is performed first on dataset $x$ (e.g., 👕 POG) and then on $y$ (e.g., ▦ PENS), we record the single-dataset performance ranks of $x$ and $y$ (e.g., Rank-4 for 👕 POG, Rank-1 for ▦ PENS). We then increment the corresponding counts in the First Step and Second Step categories. For example, a count of 33 for Rank-1 in the Second Step indicates that, across 10 test sets and considering the top 5 entries from each set, the best-performing dataset from single-dataset fine-tuning was placed in the second step 33 times.

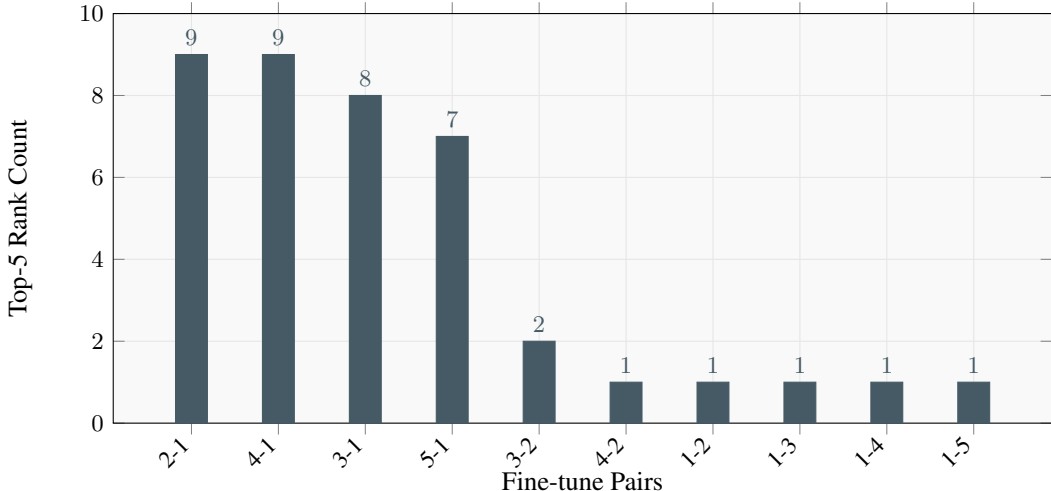

Figure 4: Impact of fine-tuning dataset order in multi-domain fine-tuning. The statistics are derived from Table 12. For each test set, we select the top-5 AUC scores from the 25 multi-domain combinations (excluding single-dataset baselines). For each top-5 result (e.g., 0.6687 on ▦ MIND), where fine-tuning is performed first on dataset $x$ (e.g., 👕 POG) and then on $y$ (e.g., ▦ PENS), we record the single-dataset performance ranks of $x$ and $y$ (e.g., Rank-4 for 👕 POG, Rank-1 for ▦ PENS). We then increment the corresponding counts for such combination. For example, the count of 9 in the 4-1 pair means that, out of all top-5 results across 10 test sets, there were 9 cases where the fourth-ranked dataset (in single-dataset fine-tuning) was used first, followed by the best-performing dataset second.

Table 12: **Performance comparison across various fine-tuning datasets and orders**. Each sub-table presents 25 AUC scores on a test set. For the entry at row $i$, column $j$ (e.g., $i = 1$, $j = 2$ in the 🔖 MIND sub-table), the model is first fine-tuned on 👕 POG, then on 🔖 PENS, yielding an AUC of 0.6687 when tested on 🔖 MIND. "Overall" indicates the average performance across the corresponding five test sets. The diagonal cells (highlighted in grey) represent results of single-dataset fine-tuning. We rank these five values and annotate the rank next to each dataset name in the column header (e.g., 🔖 PENS (1) in the 🔖 MIND sub-table indicates that 0.6823 is the highest result in single-dataset fine-tuning). For each pair of datasets, different fine-tuning orders generally yield significantly different results. The superior result for each pair is highlighted in green. The top five results among the 25 entries within each sub-table are annotated with their respective rankings.

### 👕 H&M

| | POG (4) | PENS (1) | Netflix (3) | Books (5) | Last.fm (2) |
|---|---|---|---|---|---|
| 👕 POG | 0.5922 | 0.7287 | 0.6722 | 0.5824 | 0.7496 (4) |
| 🔖 PENS | 0.6827 | 0.7517 (2) | 0.7338 | 0.6036 | 0.7355 |
| 📱 Netflix | 0.7099 | 0.7439 | 0.6655 | 0.6442 | 0.7502 (3) |
| 📖 Books | 0.7356 | 0.7450 (5) | 0.6923 | 0.5750 | 0.7444 |
| 🎵 Last.fm | 0.7367 | 0.7592 (1) | 0.7143 | 0.6096 | 0.7444 |

### 🔖 MIND

| | POG (4) | PENS (1) | Netflix (5) | Books (3) | Last.fm (2) |
|---|---|---|---|---|---|
| 👕 POG | 0.4912 | 0.6687 (5) | 0.5221 | 0.4863 | 0.5901 |
| 🔖 PENS | 0.6636 | 0.6823 (1) | 0.6186 | 0.5270 | 0.6482 |
| 📱 Netflix | 0.5530 | 0.6757 (3) | 0.4844 | 0.4992 | 0.5866 |
| 📖 Books | 0.5248 | 0.6759 (2) | 0.5223 | 0.5040 | 0.5785 |
| 🎵 Last.fm | 0.5440 | 0.6738 (4) | 0.5039 | 0.5149 | 0.5807 |

### 🔖 Micro.

| | POG (5) | PENS (1) | Netflix (3) | Books (4) | Last.fm (2) |
|---|---|---|---|---|---|
| 👕 POG | 0.5175 | 0.6443 | 0.5753 | 0.5304 | 0.6495 |
| 🔖 PENS | 0.6262 | 0.6675 | 0.6157 | 0.5508 | 0.6439 |
| 📱 Netflix | 0.6222 | 0.6693 (2) | 0.5634 | 0.5650 | 0.6688 (3) |
| 📖 Books | 0.6328 | 0.6662 (4) | 0.5784 | 0.5322 | 0.6462 |
| 🎵 Last.fm | 0.6363 | 0.6779 (1) | 0.5875 | 0.5338 | 0.6567 (5) |

### 📖 Good.

| | POG (5) | PENS (4) | Netflix (3) | Books (1) | Last.fm (2) |
|---|---|---|---|---|---|
| 👕 POG | 0.5068 | 0.5659 | 0.5631 | 0.5700 | 0.5858 |
| 🔖 PENS | 0.5632 | 0.5670 | 0.5810 | 0.6024 (1) | 0.5375 |
| 📱 Netflix | 0.5690 | 0.5721 | 0.5694 | 0.5973 (2) | 0.5743 |
| 📖 Books | 0.5822 | 0.5947 (5) | 0.5950 (4) | 0.5876 | 0.5864 |
| 🎵 Last.fm | 0.5540 | 0.5728 | 0.5748 | 0.5953 (3) | 0.5871 |

### 🎵 CDs

| | POG (5) | PENS (3) | Netflix (2) | Books (1) | Last.fm (4) |
|---|---|---|---|---|---|
| 👕 POG | 0.5011 | 0.5225 | 0.5254 | 0.5517 | 0.4947 |
| 🔖 PENS | 0.5140 | 0.5191 | 0.5366 | 0.5551 (5) | 0.5023 |
| 📱 Netflix | 0.5237 | 0.5293 | 0.5355 | 0.5602 (2) | 0.5098 |
| 📖 Books | 0.5563 (4) | 0.5340 | 0.5387 | 0.5600 (3) | 0.5039 |
| 🎵 Last.fm | 0.5098 | 0.5263 | 0.5259 | 0.5654 (1) | 0.5045 |

### Overall

| | POG (5) | PENS (1) | Netflix (3) | Books (4) | Last.fm (2) |
|---|---|---|---|---|---|
| 👕 POG | 0.5218 | 0.6260 (5) | 0.5716 | 0.5442 | 0.6139 |
| 🔖 PENS | 0.6099 | 0.6375 (4) | 0.6171 | 0.5678 | 0.6135 |
| 📱 Netflix | 0.5956 | 0.6381 (3) | 0.5636 | 0.5732 | 0.6179 |
| 📖 Books | 0.6063 | 0.6432 (1) | 0.5853 | 0.5518 | 0.6119 |
| 🎵 Last.fm | 0.5962 | 0.6420 (2) | 0.5813 | 0.5638 | 0.6147 |

### 🎬 Movie.

| | POG (5) | PENS (4) | Netflix (1) | Books (2) | Last.fm (3) |
|---|---|---|---|---|---|
| 👕 POG | 0.5354 | 0.6216 | 0.7460 (3) | 0.6649 | 0.6702 |
| 🔖 PENS | 0.5890 | 0.6311 | 0.7486 (2) | 0.6929 | 0.5829 |
| 📱 Netflix | 0.7054 | 0.7012 | 0.7422 (5) | 0.7144 | 0.7110 |
| 📖 Books | 0.7061 | 0.6832 | 0.7458 (4) | 0.6935 | 0.6843 |
| 🎵 Last.fm | 0.6406 | 0.6688 | 0.7487 (1) | 0.6737 | 0.6736 |

### 🍽 Yelp

| | POG (2) | PENS (3) | Netflix (4) | Books (1) | Last.fm (5) |
|---|---|---|---|---|---|
| 👕 POG | 0.5147 | 0.4937 | 0.5192 | 0.5660 (5) | 0.4927 |
| 🔖 PENS | 0.5295 | 0.5134 | 0.5186 | 0.5859 (1) | 0.5077 |
| 📱 Netflix | 0.5130 | 0.5138 | 0.4958 | 0.5725 (4) | 0.4825 |
| 📖 Books | 0.5597 | 0.5249 | 0.5062 | 0.5727 (3) | 0.5089 |
| 🎵 Last.fm | 0.5133 | 0.5218 | 0.4950 | 0.5754 (2) | 0.4818 |

### 🎮 Steam

| | POG (5) | PENS (2) | Netflix (3) | Books (4) | Last.fm (1) |
|---|---|---|---|---|---|
| 👕 POG | 0.5684 | 0.8902 | 0.8679 | 0.6110 | 0.9299 (3) |
| 🔖 PENS | 0.8452 | 0.8979 | 0.8781 | 0.7137 | 0.8827 |
| 📱 Netflix | 0.8826 | 0.9076 | 0.8547 | 0.7431 | 0.9297 (4) |
| 📖 Books | 0.7833 | 0.9043 | 0.8674 | 0.6373 | 0.9307 (2) |
| 🎵 Last.fm | 0.8633 | 0.9191 (5) | 0.8899 | 0.6743 | 0.9363 (1) |

### 🖥 Elec.

| | POG (5) | PENS (4) | Netflix (2) | Books (1) | Last.fm (3) |
|---|---|---|---|---|---|
| 👕 POG | 0.4896 | 0.5255 | 0.5720 | 0.6274 (4) | 0.5340 |
| 🔖 PENS | 0.5426 | 0.5353 | 0.5885 | 0.6380 (2) | 0.5211 |
| 📱 Netflix | 0.5559 | 0.5468 | 0.5947 | 0.6419 (1) | 0.5462 |
| 📖 Books | 0.5741 | 0.5564 | 0.6078 | 0.6267 (5) | 0.5506 |
| 🎵 Last.fm | 0.5445 | 0.5290 | 0.5918 | 0.6294 (3) | 0.5414 |

### 🏠 Hotel.

| | POG (3) | PENS (5) | Netflix (2) | Books (1) | Last.fm (4) |
|---|---|---|---|---|---|
| 👕 POG | 0.5142 | 0.4903 | 0.5947 | 0.6004 | 0.4990 |
| 🔖 PENS | 0.5006 | 0.4860 | 0.5892 | 0.6156 (2) | 0.5040 |
| 📱 Netflix | 0.5399 | 0.5188 | 0.6104 (4) | 0.6204 (1) | 0.5189 |
| 📖 Books | 0.5622 | 0.5033 | 0.6006 | 0.6149 (3) | 0.5216 |
| 🎵 Last.fm | 0.4921 | 0.4875 | 0.5903 | 0.6055 (5) | 0.5125 |

### Overall

| | POG (5) | PENS (4) | Netflix (1) | Books (3) | Last.fm (2) |
|---|---|---|---|---|---|
| 👕 POG | 0.5245 | 0.6043 | 0.6600 | 0.6139 | 0.6252 |
| 🔖 PENS | 0.6092 | 0.6127 | 0.6646 (2) | 0.6492 | 0.5997 |
| 📱 Netflix | 0.6394 | 0.6376 | 0.6596 (4) | 0.6585 | 0.6377 |
| 📖 Books | 0.6371 | 0.6344 | 0.6656 (1) | 0.6290 | 0.6392 (5) |
| 🎵 Last.fm | 0.6108 | 0.6252 | 0.6631 (3) | 0.6317 | 0.6291 |

**First**, in most cases, the value at row $i$, column $j$ exceeds that of the corresponding single-dataset fine-tuning result in row $j$, column $j$, indicating that using two datasets generally provides greater benefit than using only one. However, it does not necessarily surpass the value at row $i$, column $i$, since knowledge learned from dataset $i$, the first step, may be subject to catastrophic forgetting during continual fine-tuning with dataset $j$.

**Second**, building on this observation, the dataset used in the later stage (second step) of fine-tuning tends to have a dominant influence on the final performance. For example, in columns corresponding to datasets that achieved the best single-dataset results, green highlights are commonly observed (e.g.,

the 📖 PENS column when the test set is 👕 H&M, or the 📖 Books column when the test set is 📖 Good.). To further investigate this effect, we present a more detailed analysis in Figure 3, showing that datasets with stronger single-dataset performance are generally more effective when used in the second fine-tuning step.

**Third**, we further investigate which fine-tuning combinations are most likely to yield Top-5 performance. We hypothesize that this is related to the single-dataset performance of the fine-tune sets. To examine this, we present Figure 4. The results suggest that using a lower-ranked dataset in the first step, followed by the top-performing dataset in the second step, tends to produce the best outcomes for the target test set.

## C.7 ADDITIONAL EVALUATION METRICS

In the main text, we report only the AUC metric due to the space constraints. Here, we provide additional evaluation metrics, including nDCG@1, nDCG@5, MRR, Recall@1, and Recall@5, for a more comprehensive comparison.

As shown in Table 13, Table 14, and Table 15, other metrics generally align with the AUC results, supporting the consistency of our findings. We will release the complete experimental results on our website.

Table 13: Performance comparison in multi-domain recommendation scenario, with the evaluation metrics. Experiments are conducted on the 📖 MIND and 📺 Micro. datasets. We bold the best results for each metric.

| First Step | Second Step | AUC | nDCG@1 | nDCG@5 | MRR | Recall@1 | Recall@5 |
|---|---|---|---|---|---|---|---|
| | | | 📖 MIND | | | | |
| 👕 POG | 📖 PENS | 0.6687 | 0.5179 | 0.5767 | 0.5687 | 0.1865 | 0.5808 |
| 📖 PENS | 👕 POG | 0.6636 | 0.4905 | 0.5653 | 0.5513 | 0.1653 | 0.5643 |
| 👕 POG | 📺 Netflix | 0.5221 | 0.3234 | 0.4072 | 0.4333 | 0.1100 | 0.4249 |
| 📺 Netflix | 👕 POG | 0.5530 | 0.3461 | 0.4401 | 0.4607 | 0.1232 | 0.4711 |
| 👕 POG | 📘 Books | 0.4863 | 0.2675 | 0.3636 | 0.4009 | 0.0866 | 0.3944 |
| 📘 Books | 👕 POG | 0.5248 | 0.3154 | 0.4090 | 0.4330 | 0.1001 | 0.4365 |
| 👕 POG | 🎵 Last.fm | 0.5901 | 0.4206 | 0.4918 | 0.5019 | 0.1514 | 0.5071 |
| 🎵 Last.fm | 👕 POG | 0.5440 | 0.3506 | 0.4377 | 0.4562 | 0.1170 | 0.4644 |
| 📖 PENS | 📘 Books | 0.5270 | 0.2999 | 0.3987 | 0.4309 | 0.1042 | 0.4360 |
| 📘 Books | 📖 PENS | **0.6759** | **0.5342** | 0.5811 | **0.5752** | **0.1903** | 0.5777 |
| 📖 PENS | 📺 Netflix | 0.6186 | 0.4205 | 0.5067 | 0.5110 | 0.1451 | 0.5228 |
| 📺 Netflix | 📖 PENS | 0.6757 | 0.5208 | **0.5813** | 0.5743 | 0.1847 | **0.5886** |
| 📖 PENS | 🎵 Last.fm | 0.6482 | 0.4630 | 0.5440 | 0.5319 | 0.1539 | 0.5597 |
| 🎵 Last.fm | 📖 PENS | 0.6738 | 0.5182 | 0.5777 | 0.5719 | 0.1813 | 0.5814 |
| 📘 Books | 📺 Netflix | 0.5223 | 0.2990 | 0.4026 | 0.4289 | 0.0973 | 0.4326 |
| 📺 Netflix | 📘 Books | 0.4992 | 0.2688 | 0.3774 | 0.4150 | 0.0904 | 0.4128 |
| 📘 Books | 🎵 Last.fm | 0.5785 | 0.3932 | 0.4759 | 0.4878 | 0.1373 | 0.5025 |
| 🎵 Last.fm | 📘 Books | 0.5149 | 0.3005 | 0.3941 | 0.4294 | 0.1057 | 0.4276 |
| 📺 Netflix | 🎵 Last.fm | 0.5866 | 0.4026 | 0.4821 | 0.4958 | 0.1428 | 0.5032 |
| 🎵 Last.fm | 📺 Netflix | 0.5039 | 0.3012 | 0.3857 | 0.4249 | 0.1013 | 0.4178 |
| | | | 📺 Micro. | | | | |
| 👕 POG | 📖 PENS | 0.6443 | 0.6782 | 0.6782 | 0.7648 | 0.3301 | 1.0000 |
| 📖 PENS | 👕 POG | 0.6262 | 0.6638 | 0.8450 | 0.7433 | 0.3107 | 1.0000 |
| 👕 POG | 📺 Netflix | 0.5753 | 0.5945 | 0.8201 | 0.7204 | 0.2873 | 1.0000 |
| 📺 Netflix | 👕 POG | 0.6222 | 0.6665 | 0.8441 | 0.7554 | 0.3292 | 1.0000 |
| 👕 POG | 📘 Books | 0.5304 | 0.5257 | 0.7972 | 0.6889 | 0.2511 | 1.0000 |
| 📘 Books | 👕 POG | 0.6328 | 0.6635 | 0.8463 | 0.7607 | 0.3276 | 1.0000 |
| 👕 POG | 🎵 Last.fm | 0.6495 | 0.6867 | 0.6867 | 0.7749 | 0.3423 | 1.0000 |
| 🎵 Last.fm | 👕 POG | 0.6363 | 0.6705 | 0.8484 | 0.7622 | 0.3307 | 1.0000 |
| 📖 PENS | 📘 Books | 0.5508 | 0.5653 | 0.8096 | 0.7094 | 0.2780 | 1.0000 |
| 📘 Books | 📖 PENS | 0.6662 | 0.7094 | 0.8628 | 0.7828 | 0.3496 | 1.0000 |
| 📖 PENS | 📺 Netflix | 0.6157 | 0.6423 | 0.8387 | 0.7473 | 0.3137 | 1.0000 |
| 📺 Netflix | 📖 PENS | 0.6693 | 0.7028 | 0.8624 | 0.7850 | 0.3483 | 1.0000 |
| 📖 PENS | 🎵 Last.fm | 0.6439 | 0.6742 | 0.8511 | 0.7619 | 0.3258 | 1.0000 |
| 🎵 Last.fm | 📖 PENS | **0.6779** | **0.7162** | **0.8671** | **0.7921** | **0.3550** | 1.0000 |
| 📘 Books | 📺 Netflix | 0.5784 | 0.5939 | 0.8211 | 0.7248 | 0.2913 | 1.0000 |
| 📺 Netflix | 📘 Books | 0.5650 | 0.5841 | 0.8161 | 0.7192 | 0.2868 | 1.0000 |
| 📘 Books | 🎵 Last.fm | 0.6462 | 0.6879 | 0.8540 | 0.7722 | 0.3413 | 1.0000 |
| 🎵 Last.fm | 📘 Books | 0.5338 | 0.5435 | 0.8011 | 0.6999 | 0.2687 | 1.0000 |
| 📺 Netflix | 🎵 Last.fm | 0.6688 | 0.7104 | 0.8636 | 0.7873 | 0.3534 | 1.0000 |
| 🎵 Last.fm | 📺 Netflix | 0.5875 | 0.6080 | 0.8251 | 0.7334 | 0.3019 | 1.0000 |

Table 14: Performance comparison in multi-domain recommendation scenario, with the evaluation metrics. Experiments are conducted on the 📺 Micro. and ✂ Yelp datasets. We bold the best results for each metric.

| First Step | Second Step | AUC | nDCG@1 | nDCG@5 | MRR | Recall@1 | Recall@5 |
|---|---|---|---|---|---|---|---|
| 🎞 Movie. | | | | | | | |
| 👕 POG | 📖 PENS | 0.6216 | 0.4486 | 0.5870 | 0.5467 | 0.2146 | 0.7081 |
| 📖 PENS | 👕 POG | 0.5890 | 0.3970 | 0.5600 | 0.5149 | 0.1873 | 0.6875 |
| 👕 POG | 📺 Netflix | 0.7460 | 0.5643 | 0.7022 | 0.6516 | 0.2757 | 0.8207 |
| 📺 Netflix | 👕 POG | 0.7054 | 0.5360 | 0.6671 | 0.6224 | 0.2648 | 0.7896 |
| 👕 POG | 📖 Books | 0.6649 | 0.4753 | 0.6223 | 0.5757 | 0.2297 | 0.7523 |
| 📖 Books | 👕 POG | 0.7061 | 0.5277 | 0.6685 | 0.6181 | 0.2592 | 0.7952 |
| 👕 POG | 🎵 Last.fm | 0.6702 | 0.5270 | 0.6434 | 0.6072 | 0.2610 | 0.7577 |
| 🎵 Last.fm | 👕 POG | 0.6406 | 0.4749 | 0.6085 | 0.5677 | 0.2337 | 0.7323 |
| 📖 PENS | 📖 Books | 0.6929 | 0.5158 | 0.6487 | 0.6072 | 0.2552 | 0.7728 |
| 📖 Books | 📖 PENS | 0.6832 | 0.5330 | 0.6531 | 0.6079 | 0.2594 | 0.7668 |
| 📖 PENS | 📺 Netflix | 0.7486 | 0.5926 | 0.7075 | 0.6648 | 0.2934 | 0.8187 |
| 📺 Netflix | 📖 PENS | 0.7012 | 0.5546 | 0.6680 | 0.6262 | 0.2732 | 0.7790 |
| 📖 PENS | 🎵 Last.fm | 0.5829 | 0.3782 | 0.5432 | 0.4812 | 0.1570 | 0.6479 |
| 🎵 Last.fm | 📖 PENS | 0.6688 | 0.5217 | 0.6417 | 0.6024 | 0.2559 | 0.7530 |
| 📖 Books | 📺 Netflix | 0.7458 | 0.5815 | 0.7041 | 0.6576 | 0.2861 | 0.8183 |
| 📺 Netflix | 📖 Books | 0.7144 | 0.5544 | 0.6732 | 0.6279 | 0.2723 | 0.7901 |
| 📖 Books | 🎵 Last.fm | 0.6843 | 0.5417 | 0.6553 | 0.6196 | 0.2690 | 0.7688 |
| 🎵 Last.fm | 📖 Books | 0.6737 | 0.4782 | 0.6307 | 0.5871 | 0.2357 | 0.7659 |
| 📺 Netflix | 🎵 Last.fm | 0.7110 | 0.5660 | 0.6785 | 0.6417 | 0.2823 | 0.7916 |
| 🎵 Last.fm | 📺 Netflix | **0.7487** | **0.5902** | **0.7092** | **0.6657** | **0.2934** | **0.8239** |
| ✂ Yelp | | | | | | | |
| 👕 POG | 📖 PENS | 0.4937 | 0.4761 | 0.7172 | 0.6378 | 0.2227 | 0.8871 |
| 📖 PENS | 👕 POG | 0.5295 | 0.5086 | 0.7383 | 0.6575 | 0.2385 | 0.9011 |
| 👕 POG | 📺 Netflix | 0.5192 | 0.5164 | 0.7356 | 0.6576 | 0.2461 | 0.8985 |
| 📺 Netflix | 👕 POG | 0.5130 | 0.4964 | 0.7289 | 0.6572 | 0.2445 | 0.8998 |
| 👕 POG | 📖 Books | 0.5660 | 0.5535 | 0.7599 | 0.6842 | 0.2684 | 0.9166 |
| 📖 Books | 👕 POG | 0.5597 | 0.5421 | 0.7553 | 0.6813 | 0.2645 | 0.9152 |
| 👕 POG | 🎵 Last.fm | 0.4927 | 0.4765 | 0.7168 | 0.6465 | 0.2360 | 0.8917 |
| 🎵 Last.fm | 👕 POG | 0.5133 | 0.5043 | 0.7281 | 0.6559 | 0.2451 | 0.8938 |
| 📖 PENS | 📖 Books | 0.5859 | 0.5732 | 0.7695 | 0.6981 | 0.2815 | 0.9208 |
| 📖 Books | 📖 PENS | 0.5249 | 0.5097 | 0.7358 | 0.6605 | 0.2461 | 0.9022 |
| 📖 PENS | 📺 Netflix | 0.5186 | 0.5078 | 0.7340 | 0.6587 | 0.2476 | 0.9009 |
| 📺 Netflix | 📖 PENS | 0.5138 | 0.5029 | 0.7303 | 0.6552 | 0.2437 | 0.9002 |
| 📖 PENS | 🎵 Last.fm | 0.5077 | 0.4995 | 0.7262 | 0.6482 | 0.2365 | 0.8930 |
| 🎵 Last.fm | 📖 PENS | 0.5218 | 0.5019 | 0.7321 | 0.6595 | 0.2441 | 0.9003 |
| 📖 Books | 📺 Netflix | 0.5062 | 0.5105 | 0.7294 | 0.6543 | 0.2467 | 0.8953 |
| 📺 Netflix | 📖 Books | 0.5725 | **0.5635** | 0.7640 | 0.6909 | 0.2752 | 0.9175 |
| 📖 Books | 🎵 Last.fm | 0.5089 | 0.4963 | 0.7266 | 0.6568 | 0.2453 | 0.8972 |
| 🎵 Last.fm | 📖 Books | **0.5754** | 0.5628 | **0.7657** | **0.6939** | **0.2770** | **0.9220** |
| 📺 Netflix | 🎵 Last.fm | 0.4825 | 0.4684 | 0.7091 | 0.6412 | 0.2320 | 0.8850 |
| 🎵 Last.fm | 📺 Netflix | 0.4950 | 0.4885 | 0.7211 | 0.6491 | 0.2412 | 0.8955 |

Table 15: Performance comparison in multi-domain recommendation scenario, with the evaluation metrics. Experiments are conducted on the 📖 Good. and 🎵 CDs datasets. We bold the best results for each metric.

| First Step | Second Step | AUC | nDCG@1 | nDCG@5 | MRR | Recall@1 | Recall@5 |
|---|---|---|---|---|---|---|---|
| | | 📖 Good. | | | | | |
| 👕 POG | 📖 PENS | 0.5659 | 0.2801 | 0.4068 | 0.4011 | 0.1319 | 0.5098 |
| 📖 PENS | 👕 POG | 0.5632 | 0.2638 | 0.4017 | 0.3902 | 0.1224 | 0.5006 |
| 👕 POG | 📺 Netflix | 0.5631 | 0.2773 | 0.4025 | 0.3992 | 0.1322 | 0.5060 |
| 📺 Netflix | 👕 POG | 0.5690 | 0.2705 | 0.4091 | 0.4026 | 0.1316 | 0.5236 |
| 👕 POG | 📖 Books | 0.5700 | 0.2913 | 0.4137 | 0.4081 | 0.1391 | 0.5173 |
| 📖 Books | 👕 POG | 0.5822 | 0.3072 | 0.4315 | 0.4214 | 0.1483 | 0.5418 |
| 👕 POG | 🎵 Last.fm | 0.5858 | 0.2866 | 0.4311 | 0.4247 | 0.1420 | 0.5541 |
| 🎵 Last.fm | 👕 POG | 0.5540 | 0.2752 | 0.3944 | 0.3972 | 0.1339 | 0.4980 |
| 📖 PENS | 📖 Books | 0.6024 | 0.3263 | 0.4463 | 0.4380 | 0.1604 | 0.5559 |
| 📖 Books | 📖 PENS | 0.5947 | 0.3389 | 0.4488 | 0.4387 | 0.1619 | 0.5490 |
| 📖 PENS | 📺 Netflix | 0.5810 | 0.3013 | 0.4289 | 0.4209 | 0.1475 | 0.5383 |
| 📺 Netflix | 📖 PENS | 0.5721 | 0.2969 | 0.4172 | 0.4155 | 0.1434 | 0.5228 |
| 📖 PENS | 🎵 Last.fm | 0.5375 | 0.2373 | 0.3741 | 0.3605 | 0.0994 | 0.4657 |
| 🎵 Last.fm | 📖 PENS | 0.5728 | 0.2925 | 0.4182 | 0.4143 | 0.1408 | 0.5236 |
| 📖 Books | 📺 Netflix | 0.5950 | 0.3267 | 0.4455 | 0.4354 | 0.1596 | 0.5504 |
| 📺 Netflix | 📖 Books | **0.5973** | **0.3559** | **0.4529** | **0.4440** | **0.1743** | 0.5487 |
| 📖 Books | 🎵 Last.fm | 0.5864 | 0.3162 | 0.4433 | 0.4342 | 0.1567 | 0.5616 |
| 🎵 Last.fm | 📖 Books | 0.5953 | 0.3181 | 0.4452 | 0.4346 | 0.1552 | **0.5588** |
| 📺 Netflix | 🎵 Last.fm | 0.5743 | 0.2949 | 0.4227 | 0.4212 | 0.1463 | 0.5397 |
| 🎵 Last.fm | 📺 Netflix | 0.5748 | 0.3099 | 0.4234 | 0.4224 | 0.1526 | 0.5294 |
| | | 🎵 CDs | | | | | |
| 👕 POG | 📖 PENS | 0.5225 | 0.5763 | 0.7969 | 0.7180 | 0.2760 | 0.9523 |
| 📖 PENS | 👕 POG | 0.5140 | 0.5798 | 0.7964 | 0.7105 | 0.2706 | 0.9515 |
| 👕 POG | 📺 Netflix | 0.5254 | 0.5911 | 0.8011 | 0.7250 | 0.2865 | 0.9537 |
| 📺 Netflix | 👕 POG | 0.5237 | 0.5805 | 0.7978 | 0.7238 | 0.2861 | 0.9544 |
| 👕 POG | 📖 Books | 0.5517 | 0.6209 | 0.8137 | 0.7394 | 0.3027 | 0.9584 |
| 📖 Books | 👕 POG | 0.5563 | 0.6182 | 0.8126 | 0.7418 | 0.3030 | 0.9557 |
| 👕 POG | 🎵 Last.fm | 0.4947 | 0.5583 | 0.7864 | 0.7129 | 0.2763 | 0.9495 |
| 🎵 Last.fm | 👕 POG | 0.5098 | 0.5785 | 0.7941 | 0.7184 | 0.2840 | 0.9515 |
| 📖 PENS | 📖 Books | 0.5551 | 0.6242 | 0.8156 | 0.7443 | 0.3078 | 0.9591 |
| 📖 Books | 📖 PENS | 0.5340 | 0.5878 | 0.8029 | 0.7276 | 0.2854 | 0.9553 |
| 📖 PENS | 📺 Netflix | 0.5366 | 0.5969 | 0.8038 | 0.7326 | 0.2941 | 0.9534 |
| 📺 Netflix | 📖 PENS | 0.5293 | 0.5829 | 0.8001 | 0.7259 | 0.2845 | 0.9548 |
| 📖 PENS | 🎵 Last.fm | 0.5023 | 0.5619 | 0.7910 | 0.7171 | 0.2756 | 0.9511 |
| 🎵 Last.fm | 📖 PENS | 0.5263 | 0.5866 | 0.7974 | 0.7258 | 0.2879 | 0.9497 |
| 📖 Books | 📺 Netflix | 0.5387 | 0.6029 | 0.8064 | 0.7337 | 0.2948 | 0.9555 |
| 📺 Netflix | 📖 Books | 0.5602 | **0.6296** | 0.8170 | 0.7453 | 0.3100 | 0.9590 |
| 📖 Books | 🎵 Last.fm | 0.5039 | 0.5656 | 0.7886 | 0.7174 | 0.2809 | 0.9477 |
| 🎵 Last.fm | 📖 Books | **0.5654** | 0.6278 | **0.8190** | **0.7492** | **0.3114** | **0.9625** |
| 📺 Netflix | 🎵 Last.fm | 0.5098 | 0.5785 | 0.7941 | 0.7184 | 0.2840 | 0.9515 |
| 🎵 Last.fm | 📺 Netflix | 0.5259 | 0.5927 | 0.8001 | 0.7298 | 0.2942 | 0.9522 |

