# OpenReview forum: "Assessing Large Language Models (LLMs) as Foundational Recommenders: A Multi-Domain, Multi-Dataset Benchmark"
_ICLR.cc/2026/Conference — Submitted to ICLR 2026_

### Official Review · Reviewer_NfvP · 2025-10-30

**Soundness:** 2
**Presentation:** 3
**Contribution:** 2
**Rating:** 4
**Confidence:** 4

**Summary:**

This paper investigates whether large language models (LLMs), once trained on recommendation tasks, can generalize across different domains. To enable systematic evaluation, the authors propose RECBENCH-MD, a comprehensive multi-domain benchmark that integrates existing datasets from various recommendation domains. Using this benchmark, they assess the performance of 21 state-of-the-art LLMs under zero-shot, fine-tuning, and cross-domain transfer settings. The results demonstrate that in-domain fine-tuning yields the best performance, while cross-domain transfer provides practical benefits in new recommendation contexts.

**Strengths:**

This paper investigates whether large language models (LLMs), once trained on recommendation tasks, can generalize across different domains. To enable systematic evaluation, the authors propose RECBENCH-MD, a comprehensive multi-domain benchmark that integrates existing datasets from various recommendation domains. Using this benchmark, they assess the performance of 21 state-of-the-art LLMs under zero-shot, fine-tuning, and cross-domain transfer settings. The results demonstrate that in-domain fine-tuning yields the best performance, while cross-domain transfer provides practical benefits in new recommendation contexts.

**Weaknesses:**

S1: **Scope**: This paper focuses on evaluating the transferability of large language models (LLMs) as foundational recommenders across domains. It is a promising research direction in the recommendation field: How to leverage different domain's dataset for better recommend and scaling.

S2: **Experimental Rigor**: The study conducts comprehensive experiments involving 21 LLMs with diverse architectures and model scales, evaluated under various dataset combinations and learning settings (zero-shot, fine-tuning, and transfer). Such extensive empirical analysis provides insights into the generalization capabilities of LLM-based recommenders.

**Questions:**

**W1. Insufficient dataset construction details**: The paper provides limited information on how the training and evaluation subsets were constructed from each dataset. It is unclear how sampling was performed or whether consistent preprocessing protocols were applied across domains. Moreover, many datasets appear to contain only title-level textual features, which may constrain the LLM’s ability to model item semantics and contextual relevance effectively.

**W2. Limited training methodologys  for generalization evaluation.**
Although the paper assess the generalization ability of LLMs in various recommendation tasks, the evaluation setup mainly relies on supervised fine-tuning (SFT) or embedding similarity-based matching. This setup does not capture the full potential of LLM adaptation. As prior work has shown [1, 2, 3], reinforcement learning (RL) plays a crucial role in aligning model outputs with recommendation objectives, and its absence weakens the validity of the generalization claims.

**W3. Narrow evaluation of embedding quality.**
The evaluation of embedding-based recommenders is confined to direct use in a dual-tower retrieval structure. However, the quality of embeddings can manifest in multiple downstream ways. For example, LLM2Rec [4] demonstrates that LLM-derived embeddings used for item embedding initialization can enhance out-of-distribution (OOD) generalization when coupled with collaborative-filtering (CF) signals. A broader analysis of embedding quality would have made the findings more robust.

**W4:Lack of deeper analytical insights.**
The experimental analysis mainly reports performance changes across datasets and training settings, without providing sufficient interpretation of the underlying causes. For instance, the authors note that single-domain cross-dataset fine-tuning is not always more effective than cross-domain fine-tuning and attribute this to “similar data distributions,” yet no empirical evidence is offered to support this claim. Similarly, while LLMs trained on PENS datasets consistently perform better on others (Table 4), no further analysis is provided—e.g., regarding user–item matrix sparsity or domain similarity metrics—which could have explained these phenomena more convincingly.


[1] Zhou, Guorui, et al. "OneRec Technical Report." arXiv preprint arXiv:2506.13695 (2025).

[2] Liu Z, Wang S, Wang X, et al. OneRec-Think: In-Text Reasoning for Generative Recommendation[J]. arXiv preprint arXiv:2510.11639, 2025.

[3] Chu, Tianzhe, et al. "Sft memorizes, rl generalizes: A comparative study of foundation model post-training." arXiv preprint arXiv:2501.17161 (2025).

[4] He, Yingzhi, et al. "Llm2rec: Large language models are powerful embedding models for sequential recommendation." Proceedings of the 31st ACM SIGKDD Conference on Knowledge Discovery and Data Mining V. 2. 2025.

---

### Official Review · Reviewer_v2Ek · 2025-10-30

**Soundness:** 3
**Presentation:** 3
**Contribution:** 2
**Rating:** 2
**Confidence:** 4

**Summary:**

This paper introduces RECBENCH-MD, a large-scale multi-domain, multi-dataset benchmark for evaluating large language models (LLMs) on recommendation tasks.
It unifies 15 datasets across 10 domains and defines eight evaluation setups covering zero-shot, in-domain, cross-dataset, and cross-domain fine-tuning.
The authors evaluate 21 backbone models (e.g., BERT, OPT, LLaMA, Qwen, GLM, GPT-3.5, E5, RecFormer) under both prompt-based and embedding-based paradigms, and propose a new metric, RRA (Reciprocal Rank Average), to quantify dataset transferability.
Experiments show that in-domain fine-tuning generally performs best, multi-domain pretraining improves generalization, and RRA better reflects robust cross-dataset benefits.

**Strengths:**

1.	Comprehensive coverage — The benchmark spans 10 domains and 15 datasets, providing rare breadth in LLM-for-Rec evaluation.
2.	Practical findings — Demonstrates how larger models benefit from multi-domain fine-tuning and identifies strong transferable datasets.

**Weaknesses:**

1.	Lack of comparison with existing LLM-based recommendation methods.
The paper only compares different LLM backbones under a fixed fine-tuning setup, without including representative LLM-based recommendation frameworks (e.g., RecGPT[1], TALLRec[2], RankGPT[3], RecFormer[4], MoLoRec[5], One-Model-for-All[6]).
As a benchmark paper, it is insufficient to evaluate only model families (BERT vs. LLaMA vs. Qwen); it must also evaluate methodological paradigms—prompt-engineering, reasoning-based, retrieval-augmented, adapter-based, or generation-based methods.
Without such baselines, the benchmark cannot convincingly guide practitioners or measure real progress in LLM-for-Rec.
2.	No analysis of different fine-tuning strategies.
The authors fix LoRA fine-tuning for all models. Yet, diverse tuning strategies (full-finetune, prefix-tuning, adapter, instruction-tuning, multi-step sequential tuning, in-context learning) often lead to dramatically different behaviors in recommendation tasks.
A benchmark aiming to be “foundational” should at least include ablations or comparative studies across these tuning paradigms to understand their trade-offs in accuracy vs. efficiency.
3.	Insufficient discussion on textual vs. non-textual datasets.
Many datasets in RECBENCH-MD differ substantially in text richness—e.g., news and books datasets contain rich natural language, while user–item logs (e.g., Amazon Electronics, Yelp) are structured or sparse.
Since LLMs rely on textual semantics, performance gaps across such dataset types are expected to be significant, yet the paper lacks a dedicated analysis or normalization.
Without explicitly separating “text-dominant” vs. “non-textual” domains, it is difficult to interpret whether the performance differences stem from model architecture or simply from text availability.

Refs:
[1] Hoang Ngo and Dat Quoc Nguyen. 2024. RecGPT: Generative Pre-training for Text-based Recommendation. In Proceedings of the 62nd Annual Meeting of the Association for Computational Linguistics (Volume 2: Short Papers), pages 302–313, Bangkok, Thailand. Association for Computational Linguistics.
[2] Keqin Bao, Jizhi Zhang, Yang Zhang, Wenjie Wang, Fuli Feng, and Xiangnan He. 2023. TALLRec: An Effective and Efficient Tuning Framework to Align Large Language Model with Recommendation. In Proceedings of the 17th ACM Conference on Recommender Systems (RecSys '23). Association for Computing Machinery, New York, NY, USA, 1007–1014.
[3] Weiwei Sun, Lingyong Yan, Xinyu Ma, Shuaiqiang Wang, Pengjie Ren, Zhumin Chen, Dawei Yin, and Zhaochun Ren. 2023. Is ChatGPT Good at Search? Investigating Large Language Models as Re-Ranking Agents. In Proceedings of the 2023 Conference on Empirical Methods in Natural Language Processing, pages 14918–14937, Singapore. Association for Computational Linguistics.
[4] Jiacheng Li, Ming Wang, Jin Li, Jinmiao Fu, Xin Shen, Jingbo Shang, and Julian McAuley. 2023. Text Is All You Need: Learning Language Representations for Sequential Recommendation. In Proceedings of the 29th ACM SIGKDD Conference on Knowledge Discovery and Data Mining (KDD '23). Association for Computing Machinery, New York, NY, USA, 1258–1267
[5] Hou, M. et al. (2025). MoLoRec: A Generalizable and Efficient Framework for LLM-Based Recommendation. arXiv:2502.08271.
[6]  Zuoli Tang, Zhaoxin Huan, Zihao Li, Xiaolu Zhang, Jun Hu, Chilin Fu, Jun Zhou, Lixin Zou, and Chenliang Li. 2025. One Model for All: Large Language Models Are Domain-Agnostic Recommendation Systems. ACM Trans. Inf. Syst. 43, 5, Article 118 (September 2025), 27 pages. https://doi.org/10.1145/3705727

**Questions:**

How do you handle datasets with little or no textual description (e.g., click or rating logs)?
Are item titles/descriptions uniformly available, and do you normalize token length or vocabulary coverage across domains?

---

### Official Review · Reviewer_TWdo · 2025-11-01

**Soundness:** 2
**Presentation:** 3
**Contribution:** 2
**Rating:** 4
**Confidence:** 4

**Summary:**

This paper presents RECBENCH-MD, a large-scale, multi-domain, and multi-dataset benchmark designed to systematically assess large language models (LLMs) as foundational recommenders. The benchmark includes 15 datasets from 10 domains and evaluates 21 state-of-the-art LLMs under various scenarios—zero-shot, fine-tuning, and cross-domain transfer. Eight comprehensive experimental settings (A–H) are defined to cover single- and multi-domain, single- and multi-dataset cases. The authors report several key findings: in-domain fine-tuning yields the best results; cross-dataset transfer offers practical benefits in new recommendation contexts; and multi-domain training improves model adaptability. The paper provides extensive experimental analyses, open-source data, and code to facilitate reproducibility.

**Strengths:**

1. Comprehensive and systematic benchmark design: RECBENCH-MD is arguably the most extensive benchmark to date for evaluating LLM-based recommenders, covering a diverse range of datasets, domains, and models.
2. Clear and well-structured experimental taxonomy: The eight well-defined evaluation settings (A–H) provide a clear taxonomy for understanding the effects of domain, dataset, and fine-tuning regime, which is useful for both researchers and practitioners.
3. High reproducibility and transparency: The authors provide detailed experimental configurations, including LoRA fine-tuning settings and dataset statistics, and commit to releasing code and checkpoints publicly, which enhances reproducibility and research value.

**Weaknesses:**

1. Limited methodological novelty: The main contribution lies in benchmark construction and empirical evaluation rather than introducing a new algorithm or theoretical insight. This makes the work primarily engineering-oriented rather than conceptual.
2. Lack of traditional baselines: The comparison focuses on LLMs and a single LLM-based recommender baseline (RecBase), but omits traditional recommendation models (e.g., SASRec, GRU4Rec, BERT4Rec) that could contextualize LLM performance, especially since LLMs may not outperform collaborative models in all scenarios.
3. Efficiency and cost analysis is superficial: The paper briefly mentions that LLMs are inefficient in fine-tuning and inference but does not provide quantitative comparisons (e.g., GPU hours, FLOPs, or memory cost) or discuss trade-offs with smaller adapters or quantization techniques.
4. Limited task coverage: The benchmark currently excludes sequential recommendation, a core task in the field. Since the authors already handle user-item interaction sequences, extending to sequential prediction should be feasible and meaningful.
5. Information leakage concern: All datasets used are public, and many LLMs were trained on web-scale corpora that could overlap with these sources (e.g., Amazon, Yelp, Goodreads). The paper does not discuss how potential data contamination or training leakage was mitigated or verified.
6. Incomplete fine-tuning exploration: The study only applies LoRA-based parameter-efficient tuning, without examining whether full fine-tuning could yield substantially different conclusions. This omission may limit the validity of claims about LLMs’ adaptation capabilities.
7. Limited analysis depth: While the experimental coverage is broad, much of the discussion remains observational (e.g., “larger models generalize better”) without deeper causal or representational analysis—for instance, quantifying domain similarity or understanding feature-level transfer.
8. Metric and statistical reporting: Although the authors state that results are averaged over five runs with significance testing (p < 0.05), no variance or confidence intervals are reported, making it difficult to assess robustness.
9. Choice of similarity metric (Equation 3): The embedding-based recommendation model uses cosine similarity, but the rationale is unclear. It would be valuable to see whether dot-product or L2 distance were tested and how sensitive results are to this choice.

**Questions:**

Please refer to Weekness.

---

### Official Review · Reviewer_WW2s · 2025-11-01

**Soundness:** 2
**Presentation:** 2
**Contribution:** 2
**Rating:** 2
**Confidence:** 4

**Summary:**

This paper focuses on the potential of large language models (LLMs) as foundational recommenders. To address the limitations of "single-dataset/single-domain" and fragmented conclusions in current LLM-based recommendation research, it proposes RECBENCH-MD, the first multi-domain, multi-dataset benchmark. Covering 15 datasets across 10 domains (e.g., e-commerce, entertainment, social media), the benchmark systematically evaluates the recommendation capabilities of 21 state-of-the-art LLMs under zero-resource, fine-tuning, and transfer-learning scenarios. By comparing prompt-based recommendation and embedding-based matching—the study reveals key patterns of LLMs in recommendation tasks (e.g., in-domain fine-tuning yields the best performance, cross-dataset transfer has practical value) and open-sources code and data to support future research.

**Strengths:**

## 1. Innovation and Comprehensiveness of the Benchmark:
Fills the gap in "multi-domain, multi-dataset evaluation of LLM recommendation capabilities". The 8 scenario designs cover the full range from basic zero-shot to complex multi-domain transfer, far exceeding existing benchmarks (e.g., LLMRec only supports single-domain single-paradigm), providing a unified evaluation standard for the field.

## 2. Practical Value and Openness:
Open-sourced resources lower the threshold for subsequent research.

**Weaknesses:**

## 1. Limited Scenario Coverage
The paper only evaluates two recommendation tasks (prompt-based ranking and embedding-based matching) but omits sequential recommendation—a core scenario in real-world systems (e.g., predicting the next item a user will interact with). This limits the benchmark’s applicability to dynamic user behavior modeling.

## 2. Neglect of LLM-Based Generative Recommendation.
The paper completely overlooks LLM-based generative recommendation—a key direction in recent research that includes text-driven generative recommendation (e.g., generating personalized item descriptions or recommendation reasons) and SID (Semantic Identifier)-based generative recommendation (represented by OneRec). By focusing solely on discriminative tasks (prompt-based ranking, embedding-based matching), the benchmark fails to capture the unique value of LLMs in generating interpretable, personalized recommendation outputs. This omission limits the benchmark’s relevance to emerging generative recommendation scenarios.

## 3. Overemphasis on Raw LLM Performance While Underplaying Practical Improvements.
The paper devotes extensive discussion to the recommendation performance of raw LLMs (e.g., zero-shot or basic fine-tuning), despite the community’s general consensus that raw LLMs perform poorly in recommendation tasks. More critically, it provides minimal analysis of practical strategies to improve LLM performance for recommendation—such as hybrid models (combining LLMs with traditional recommenders), specialized pre-training for recommendation (e.g., incorporating user-item interaction signals), lightweight adaptation (beyond LoRA), or recommendation-oriented reinforcement learning. This gap makes the research less actionable for addressing the "poor raw performance" issue.

**Questions:**

## 1. Integrate Generative Recommendation to Expand Benchmark Coverage.
To make RECBENCH-MD a truly comprehensive benchmark for LLM-based recommendation, we suggest adding generative recommendation tasks with clear standardization:(1) Task Definition: Include two typical generative tasks—"next-item title generation" (predicting next items via text) and "next-item SID generation" (predicting next items via SIDs).

This addition will enable the benchmark to capture LLMs’ unique generative advantages, which discriminative tasks alone cannot reflect.

## 2. Add a "Practical Improvement Track" for LLM Recommendation.
To address the gap between raw LLM performance and real-world needs, we recommend adding a dedicated track in RECBENCH-MD for improved LLM-based recommendation methods. This track will make the benchmark more actionable, as it directly addresses the "how to make LLMs work better for recommendation" question faced by researchers and engineers.

## 3. Considerations for Excluding Generative Recommendation Tasks.
As a multi-domain benchmark aiming to assess LLMs as "foundational recommenders," generative recommendation (e.g., SID-based generation in OneRec) has become a core capability of LLMs in recommendation. Could you elaborate on:(1) The specific reasons for excluding generative tasks in the current version?(2) Whether there is a roadmap to integrate generative tasks, and if so, how you plan to standardize them? This clarification is critical for evaluating whether RECBENCH-MD can adapt to the shift from discriminative to generative recommendation, a key trend in the field.

**Details Of Ethics Concerns:**

no concern

---

### Meta-Review · Area_Chair_UUUx · 2026-01-04

**Summary:**

The paper proposes **RecBench-MD**, a benchmark designed to evaluate Large Language Models (LLMs) as foundational recommenders across 15 datasets and 10 domains. While the reviewers acknowledged the extensive data collection effort, there was a unanimous consensus that the paper falls below the acceptance threshold for ICLR (Scores: 2, 2, 4, 4). The primary concerns driving this decision are:

1.  **Incomplete Task Coverage:** **Reviewer WW2s** strongly criticized the exclusion of "Generative Recommendation" (e.g., text or SID generation), arguing that this overlooks the unique advantage of LLMs compared to traditional methods. Additionally, both **Reviewer WW2s** and **Reviewer TWdo** noted the omission of "Sequential Recommendation," a core task in the field, which limits the benchmark's utility.
2.  **Missing Critical Baselines:** **Reviewer v2Ek** pointed out a significant flaw in the experimental design: the absence of state-of-the-art LLM-based recommendation frameworks (e.g., TALLRec, RecGPT, RankGPT). Similarly, **Reviewer TWdo** highlighted the lack of traditional strong baselines (e.g., SASRec, BERT4Rec), making it impossible to contextualize the performance of the "raw" LLMs evaluated.
3.  **Insufficient Methodological Depth:** **Reviewer v2Ek** and **Reviewer NfvP** raised concerns that the paper relies solely on LoRA fine-tuning, failing to explore other critical strategies like full fine-tuning or Reinforcement Learning.
4.  **Validity and Efficiency Concerns:** **Reviewer TWdo** raised valid concerns regarding potential data leakage (public datasets present in pre-training corpora) and the lack of computational cost analysis. **Reviewer v2Ek** also noted the failure to distinguish between text-rich and text-sparse datasets in the analysis.

**Reviewer Concerns:**

Since no rebuttal was provided, all concerns raised by the reviewers remain outstanding and unaddressed.

**Reviewer Scores:**

Given that the authors did not participate in the rebuttal phase, there is no new information to motivate a positive change in scores.

---

### Decision · Program_Chairs · 2026-01-26

Reject